# Adipocyte heterogeneity regulated by the Bithorax Complex-Wnt signaling crosstalk in *Drosophila*

Rajitha-Udakara-Sampath Hemba-Waduge [1,4], Mengmeng Liu[1,4], Xiao Li [2,4], Jasmine L Sun[1], Elisabeth A Budslick[1], Sarah E Bondos [3] & Jun-Yuan Ji [1✉]

## Abstract

**Adipocytes play essential roles in lipid metabolism and energy homeostasis, with regional differences affecting their functions and disease susceptibility. However, the mechanisms underlying this regional heterogeneity remain unclear. Here we demonstrate that the *Bithorax Complex* (*BX-C*) genes, specifically *abdominal A* (*abd-A*) and *Abdominal B* (*Abd-B*), define regional differences in *Drosophila* larval adipocytes. Abdominal adipocytes, expressing *abd-A* and *Abd-B* exhibit unique characteristics compared to thoracic adipocytes, with active Wnt/Wingless signaling further amplifying these regional differences. Depleting *abd-A* and *Abd-B* in adipocytes delays larval-pupal transition, causes pupal lethality, and attenuates the expression of Wnt/Wg target genes, thereby dampening Wnt signaling-induced lipid mobilization. Additionally, Wnt signaling enhances the transcription of *abd-A* and *Abd-B*, establishing a feedforward loop that reinforces the interplay between Wnt signaling and *BX-C* genes. These findings reveal how the cell-autonomous expression of *BX-C* genes defines adipocyte heterogeneity, a process further modulated by Wnt signaling in *Drosophila* larvae.**

**Keywords** Wnt/Wingless Signaling; The Bithorax Complex; Lipid Homeostasis; Adipocyte Heterogeneity; *Drosophila*
**Subject Categories** Development; Genetics, Gene Therapy & Genetic Disease; Metabolism

## Introduction

Adipocytes in different parts of the body exhibit distinct functions and regulatory mechanisms, with varying metabolic profiles that are essential for maintaining lipid metabolism and energy homeostasis. Disruptions in these processes are linked to diseases such as obesity, diabetes, cardiovascular diseases, and cancer (Bhaskaran et al, 2014; Desvergne et al, 2006; Lusis et al, 2008; Renehan et al,

2008; Rosen and MacDougald, 2006; Tchkonia et al, 2013). Notably, the specific locations of excess fat accumulation can pose distinct health vulnerabilities: visceral fat, located around internal organs, significantly increases the risk of diseases compared to subcutaneous fat stored in the lower body (Despres, 2012; Jensen, 2008; Tchkonia et al, 2013). Thus, it is important to understand the signaling pathways and developmental processes that regulate adipocyte heterogeneity across different regions of the body.

*Drosophila* serves as a prominent model organism for studying lipid metabolism, inter-organ communications, and disease modeling (Bier, 2005; Leopold and Perrimon, 2007). Its fat body functions analogously to mammalian adipose tissue in regulating lipid homeostasis (Arrese and Soulages, 2010; Baker and Thummel, 2007; Leopold and Perrimon, 2007). In *Drosophila* larvae, the fat body consists of a monolayer of adipocytes aligned along the anteroposterior axis (Rizki, 1978). While fundamental metabolic processes—such as glycolysis, the TCA cycle, and lipid metabolism—are conserved between *Drosophila* and mammals (Heier and Kuhnlein, 2018; Mattila and Hietakangas, 2017), key lipid metabolic events, including adipogenesis (adipocyte differentiation), lipogenesis, and lipolysis, occur at distinct developmental stages in *Drosophila*. Adipogenesis is restricted to late embryogenesis, whereas lipogenesis, lipolysis, and fatty acid β-oxidation are primarily active during the larval and pupal stages (Heier and Kuhnlein, 2018; Hoshizaki, 2005; Mattila and Hietakangas, 2017; Rizki, 1978). This temporal separation contrasts with the more intricately interconnected and continuous nature of these processes in mammals, making *Drosophila* a valuable model for independently dissecting different aspects of lipid metabolism.

Wnt signaling is well-established to negatively regulate adipogenesis in mammals (Ross et al, 2000; Waki et al, 2007), but its role in regulating lipogenesis and lipolysis remains less understood. Previous studies indicate that active Wnt signaling suppresses the expression of lipogenic enzymes in β-catenin-deficient mouse adipocytes and the liver of juvenile turbot (Bagchi et al, 2020; Liu et al, 2016). ChIP-seq analysis further reveals the enrichment of Tcf7l2/TCF4 at lipogenic genes in cultured adipocytes, suggesting direct regulation by Wnt signaling (Bagchi et al, 2020). Recent research in *Drosophila* larval adipocytes has shown that Wnt/Wingless (Wg) signaling represses genes involved in de novo

[1]Department of Biochemistry and Molecular Biology, Tulane University School of Medicine, Louisiana Cancer Research Center, 1700 Tulane Avenue, New Orleans, LA 70112, USA. [2]Lewis-Sigler Institute of Integrative Genomics, Princeton University, Princeton, NJ 08540, USA. [3]Department of Medical Physiology, College of Medicine, Texas A&M University Health Science Center, College Station, TX 77843, USA. [4]These authors contributed equally: Rajitha-Udakara-Sampath Hemba-Waduge, Mengmeng Liu, Xiao Li. ✉E-mail: ji@tulane.edu

lipogenesis, fatty acid β-oxidation, and lipid droplet-associated proteins (Liu et al, 2024). This results in the inhibition of lipogenesis while concurrently promoting lipolysis and lipid mobilization. Conversely, the downregulation of Wnt/Wg signaling increased the expression of lipid metabolism-related genes and triglyceride levels, suggesting its important role in regulating lipid metabolism in adipocytes (Liu et al, 2024).

Evolutionarily conserved homeobox (Hox) genes are crucial in establishing anteroposterior segmental differences in the body plan across animal lineages (Garcia-Fernandez, 2005; Gehring et al, 2009). In Drosophila, Hox genes are organized into two clusters: the Antennapedia Complex (ANT-C), which controls the anterior segment identity, and the Bithorax Complex (BX-C), which regulates the posterior segment identity. The BX-C cluster includes three Hox genes: Ultrabithorax (Ubx), abdominal-A (abd-A), and Abdominal-B (Abd-B). While BX-C proteins are well-known for their roles in regulating embryonic segmentation (Bender, 2020; Gehring et al, 2009; Maeda and Karch, 2006), their postembryonic roles in different tissues, particularly adipose tissue, remain less understood. Immunostaining studies have confirmed the presence of BX-C proteins in the abdominal fat body (Marchetti et al, 2003). However, their expression was reported to decline in third-instar (L3) wandering larvae, a reduction necessary for the derepression of autophagy-related genes and the activation of developmental autophagy (Banreti et al, 2014; Duffraisse et al, 2020). Despite these insights, the role of BX-C proteins in regulating lipid metabolism and adipocyte heterogeneity has remained unexplored.

Here, we report that adipocyte heterogeneity in the Drosophila larval fat body is defined by the differential expression of BX-C proteins Abd-A and Abd-B, which are abundant in abdominal, but not in thoracic adipocytes. This regional difference is further amplified by active Wnt/Wg signaling. Abd-A and Abd-B, but not Ubx, are specifically expressed in abdominal adipocytes of L3 wandering larvae, where they modulate Wnt/Wg target gene expression and lipid metabolism. Depleting BX-C genes in abdominal adipocytes delays larval-pupal transition and causes pupal lethality. Furthermore, Wnt/Wg signaling enhances the transcription of abd-A and Abd-B in abdominal adipocytes, establishing a feedforward loop that reinforces adipocyte heterogeneity along the anteroposterior body axis in Drosophila larvae.

## Results

### Regional differences in larval adipocytes correlate with the differential expression of BX-C proteins

This study was prompted by a curious observation linking fat storage to Wnt/Wg signaling in Drosophila. At the L3 wandering stage, Axin127 mutant larvae become transparent (Fig. 1B, compared to the control in Fig. 1A) and eventually die as pupae (Zhang et al, 2017). Such phenotypes are characteristic of disrupted lipid storage, which affects larval opacity. Axin (Axn) functions as a negative regulator of Wnt/Wg signaling (Hamada et al, 1999). Previous studies demonstrated that defective Axn activates Wnt/Wg signaling, leading to decreased triglyceride levels and increased free fatty acids, thereby disrupting lipid homeostasis (Liu et al, 2024; Zhang et al, 2017). Active Wnt signaling promotes lipolysis and fatty acid β-oxidation while inhibiting lipogenesis through the

transcriptional repression of key factors involved in these processes (Liu et al, 2024). Interestingly, fat accumulation defects in Axn127 mutants were exclusively restricted to the abdominal fat body, while the thoracic region remained unaffected (Fig. 1B). This striking regional difference in adipocytes has not been previously described.

Using the GAL4-UAS system to activate Wnt signaling in the larval fat body, we consistently observed similar regional differences. Depleting Axn with fat body-specific Gal4 drivers, such as SREBP-Gal4 (Fig. 1C vs. Fig. 1D; Appendix Fig. S1A vs. Appendix Fig. S1B), dCg-Gal4 (Appendix Fig. S1D vs. Appendix Fig. S1E), and r4-Gal4 (Appendix Fig. S1F vs. Appendix Fig. S1G), caused defects in the abdominal fat body, but left the thoracic region unaffected. Similarly, depleting slmb (supernumerary limbs, which encodes an E3 ubiquitin ligase responsible for degrading Armadillo/Arm, the β-catenin homolog) (Jiang and Struhl, 1998), activated Wnt signaling and disrupted lipid metabolism in abdominal adipocytes, while sparing the thoracic adipocytes mostly unaffected (Appendix Fig. S1C vs. Appendix Fig. S1A). These observations suggest that the regional differences observed are not caused by unrelated lesions on the Axn127 chromosome or by random variegations of the Gal4-UAS system.

At the cellular level, larval adipocytes from both thoracic and abdominal regions in w1118 (Fig. 1E,E') and SREBP-Gal4 heterozygous controls (Fig. 1G,G') were uniformly sized and shaped. In contrast, in Axn127 homozygotes or Axn-depleted larvae, lipid accumulation was markedly reduced in the abdominal region compared to the thoracic region (Fig. 1F' vs. Fig. 1F; Fig. 1H' vs. Fig. 1H). The activation of Wnt signaling is stronger in Axn127 larvae than in AxnRNAi larvae, resulting in a more substantial decrease in average adipocyte size in Axn127 (Appendix Fig. S2A). These effects are evident when we repeat the quantifications by considering only the Wnt-active small adipocyte population (Appendix Fig. S2B). While thoracic adipocytes remained uniform in size (Fig. 1F,H), abdominal adipocytes were predominantly smaller, with occasional large cells interspersed (Fig. 1F',H'). These observations reveal distinct responses of thoracic and abdominal adipocytes to active Wnt signaling.

The large size of larval adipocytes and their abundant lipid droplets make single-cell RNA-seq analysis difficult. To analyze the differences between thoracic and abdominal adipocytes, we performed transcriptomic analysis on dissected fat body regions from control 'dCg-Gal4/+; +' animals (Appendix Fig. S1D,H). Differential gene expression and pathway analysis identified significant differences in biological processes such as 'hydrolase,' 'plasma membrane,' 'nucleotide binding,' and 'developmental proteins' (Fig. 1I). Within the 'developmental proteins' category, 10 out of 17 genes showed significantly higher expression in the abdominal adipocytes (Fig. 1J). Five of these genes (piwi, vas, spn-E, croc, and Fas2) are expressed in gonads (Tu et al, 2020), likely reflect the presence of embedded primordial gonads in the abdominal fat body and were therefore excluded from further analyses. Additionally, depleting or overexpressing drm (drumstick), odd (odd skipped), and Wnt4 (Appendix Fig. S3) did not affect fat accumulation or adipocyte size. Wnt4 is known to antagonize Wg/Wnt1 signaling in the ventral ectoderm of Drosophila embryos and suppress Wg-induced axis formation in Xenopus embryos (Gieseler et al, 1999). Therefore, these genes were also excluded from further analyses.

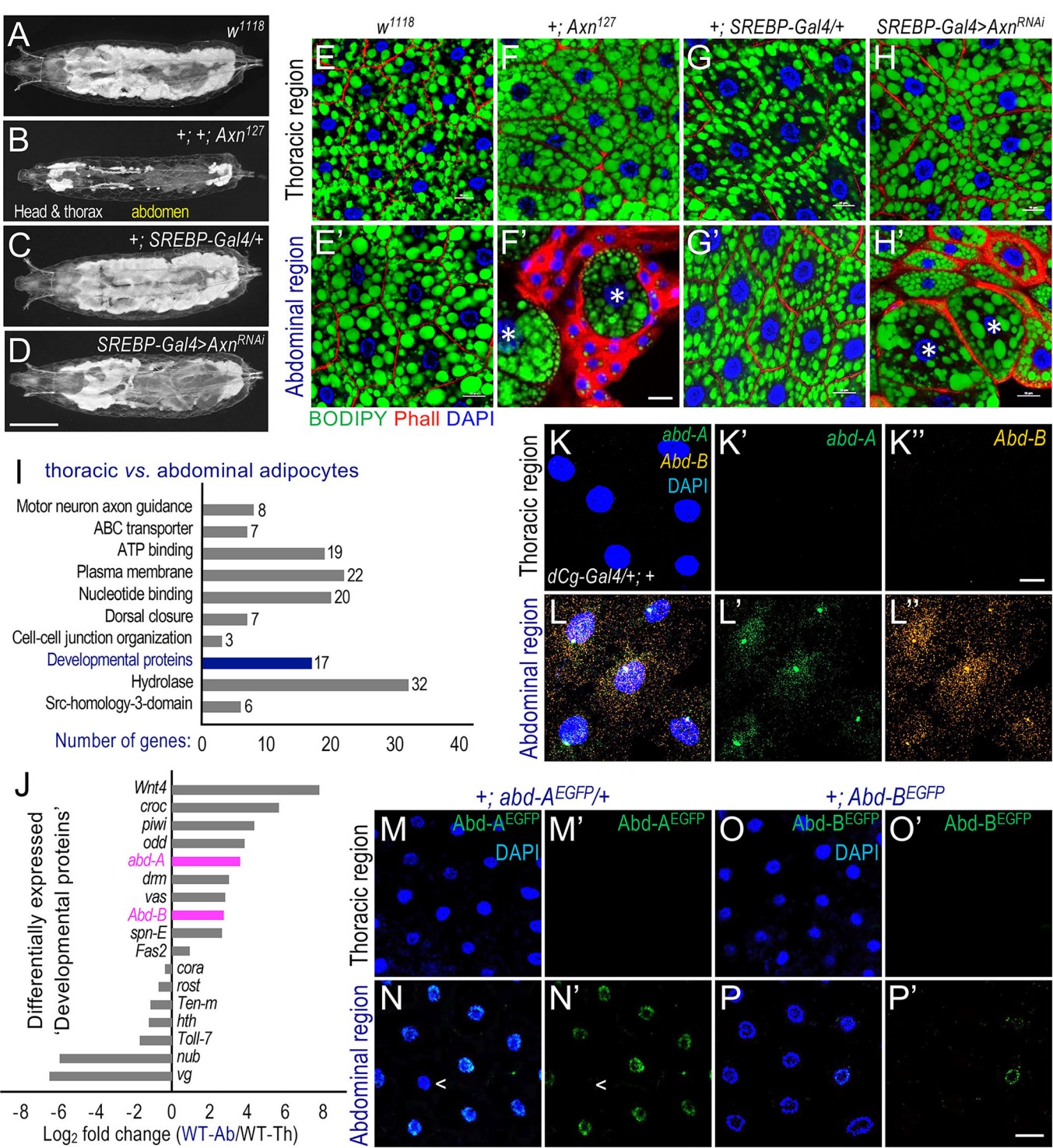

BODIPY Phall DAPI

**I** thoracic *vs.* abdominal adipocytes

Number of genes

**J** Differentially expressed 'Developmental proteins'

Log₂ fold change (WT-Ab/WT-Th)

To validate the purity of our RNA-Seq datasets, we cross-referenced our data with nine genes known to be enriched in the larval intestine, as reported in the FlyAtlas2 database (https://flyatlas.gla.ac.uk/FlyAtlas2/index.html): *CG16723*, *Muc68E*, *CG18404*, *Bace*, *CG43187*, *LManIII*, *CG6996*, *CG43348*, and *Mip*. As shown in Appendix Fig. S4A,B, these genes were undetectable or barely detectable in our RNA-seq analyses of dissected fat bodies due to extremely low counts. In contrast, the

expression of typical adipocyte-enriched genes, including *FASN1*, *Lsd-1*, *Lsd-2*, and *whd*, was robust and consistent, indicating minimal intestinal contamination and confirming the high purity of our samples.

Given their established roles in specifying regions along the anteroposterior axis, we focused on the homeobox genes *abd-A* and *Abd-B* (Fig. 1J), which encode two of the three members of the Bithorax Complex (BX-C) (Bender, 2020; Gehring et al, 2009;

**Figure 1. Regional differences in *Drosophila* larval fat body correlate with differential expression of *abd-A* and *Abd-B*.**

(A–D) Whole larvae images of (A) $w^{1118}$, (B) +; $Axn^{127}$, (C) +; SREBP-Gal4/+, and (D) UAS-$Axn^{RNAi}$/+; SREBP-Gal4/+, showing transparency in the abdominal region upon activation of Wnt signaling. Scale bar in (D) applies to images shown in (A–D): 1.0 mm. (E–H) Representative confocal images of larval adipocytes from indicated genotypes, stained with DAPI (blue), BODIPY (green), and Phalloidin (Phall, red). Occasional large adipocytes are marked with asterisks (*). Genotypes: (E–E') $w^{1118}$, (F–F') +; $Axn^{127}$ (G- G') +; SREBP-Gal4/+, and (H–H') UAS-$Axn^{RNAi}$/+; SREBP-Gal4/+. Scale bar in (F') applies to images shown in (E–E') to (H–H'): 10 μm. (I) Common pathways differentially regulated between the thoracic and abdominal regions of fat body dissected from 'dCg-Gal4/+;+' larvae (n = 3, independent biological repeats). (J) Expression patterns of 17 genetic factors categorized under 'Developmental proteins'. *abd-A* and *Abd-B* show significantly higher expression levels in the abdominal region ('WT-Ab') of larval fat body compared to the thoracic region ('WT-Th'). (K, L) HCR RNA-FISH imaging of *abd-A* (green) and *Abd-B* (orange) mRNA transcripts in larval adipocytes dissected from 'dCg-Gal4/+;+' control larvae. Scale bar in (K") applies to images shown in (K, L): 10 μm. (M–P) Expression pattern of endogenously EGFP-tagged Abd-A and Abd-B in the larval fat body. Genotypes: (M–M', N–N') '+; $abd$-$A^{EGFP}$/+' and (O-O', P–P') '+; $Abd$-$B^{EGFP}$'. Adipocytes that do not exhibit *abd-A* expression are marked with '<'. Scale bar in (P') applies to images shown in (M–M') to (P–P'): 20 μm. Source data are available online for this figure.

Maeda and Karch, 2006). Using multiplexed HCR RNA-FISH (hybridization chain reaction RNA fluorescence in situ hybridization) imaging technology (Choi et al, 2018), we confirmed that *abd-A* and *Abd-B* mRNAs are specifically expressed in abdominal adipocytes (Fig. 1L–L"), but not thoracic adipocytes (Fig. 1K–K") of wild-type larvae, consistent with our RNA-seq data. This spatial expression pattern is also consistent with previous reports of BX-C protein expression in the abdominal fat body (Marchetti et al, 2003). However, immunostaining for BX-C proteins in larval adipocytes has yielded inconsistent results (Banreti et al, 2014; Duffraisse et al, 2020; Marchetti et al, 2003). To overcome this limitation, we used CRISPR-generated EGFP-tagged endogenous BX-C lines, which enabled more sensitive and specific detection.

To examine Ubx expression in the fat body, we used a CRISPR-Cas9-generated EGFP-tagged Ubx line (Domsch et al, 2019). $Ubx^{EGFP}$ fusion proteins were readily detected in the haltere discs of both early L3 pre-wandering and L3 wandering larvae (Appendix Fig. S5DA–A',D–D') (Domsch et al, 2019). However, no $Ubx^{EGFP}$ signal was observed in the fat body at either stage (Appendix Fig. S5B–C',E–F'). Similarly, we detected no discernible $Ubx^{EGFP}$ expression in the ovaries during either the pre-wandering or wandering stages (Appendix Fig. S5G–H'), and only a weak signal in the testes during the pre-wandering stages (Appendix Fig. S5I–J').

To further analyze Abd-A and Abd-B expression, we used CRISPR-Cas9 to genetically tag their endogenous loci with EGFP. The design of this strategy is illustrated in Appendix Fig. S6, and both transgenic lines were validated by sequencing (see "Methods" for details). Endogenous Abd-$A^{EGFP}$ (Fig. 1M,M' vs. Fig. 1N,N') and Abd-$B^{EGFP}$ (Fig. 1O,O' vs. Fig. 1P,P') were detected in the nuclei of abdominal adipocytes but not in thoracic adipocytes. Both Abd-$A^{EGFP}$ and Abd-$B^{EGFP}$ proteins were also expressed in subsets of cells within the ovaries and testes during the pre-wandering and wandering larval stages (Appendix Fig. S7). Interestingly, expression of these proteins in abdominal adipocytes was heterogeneous, with sporadic cells lacking detectable levels (Fig. 1N',P'). Together, these observations confirm that endogenous *abd-A* and *Abd-B* are specifically expressed in abdominal, but not thoracic, adipocytes.

To analyze the role of BX-C proteins in larval adipocytes, we generated transgenic RNAi lines using the pNP vector (Qiao et al, 2018), which allows simultaneous depletion of multiple BX-C members. These transgenic RNAi lines were validated by sequencing (Appendix Fig. S8A) and further confirmed by RNA-seq analysis (see below in Fig. 3B). Simultaneous depletion of Abd-A and Abd-B in larval adipocytes delayed the larval-pupal transition

by one day, whereas individual depletion had weaker effects (Appendix Fig. S9). Depleting all three BX-C genes led to similar effects to co-depletion of Abd-A and Abd-B (Appendix Fig. S9), indicating that Ubx plays little role in this process. This observation is consistent with the undetectable levels of $Ubx^{EGFP}$ in larval adipocytes (Appendix Fig. S5). Based on these observations, we focused our subsequent analyses on the role of Abd-A and Abd-B.

## Differential expression of Abd-A and Abd-B defines region-specific Wnt regulation of lipid metabolism

The region-specific effects of Wnt signaling on lipid accumulation, along with the expression of Abd-A and Abd-B in abdominal adipocytes, prompted us to investigate their potential interplay in shaping adipocyte heterogeneity (Fig. 2A). To test whether altering Abd-A and Abd-B levels modulates lipid metabolism, potentially in synergy with Wnt signaling, we examined whether ectopic expression of *abd-A* or *Abd-B* in thoracic adipocytes, combined with *Axn* depletion, could recapitulate the lipid accumulation defects observed in abdominal adipocytes. In the control (*Axn* depletion), fat accumulation was reduced in most adipocytes, consistent with previous findings (Liu et al, 2024), with occasional large adipocytes interspersed (Fig. 2C' vs. Fig. 2B'). In contrast, thoracic adipocytes remain unaffected (Fig. 2C vs. Fig. 2B). However, ectopic expression of *abd-A* (Fig. 2E,E') or *Abd-B* (Fig. 2G,G') in the *Axn*-depleted background induced heterogeneity in thoracic adipocytes, mimicking the phenotypes of abdominal adipocytes with active Wnt signaling. In contrast, ectopic expression of *abd-A* alone (Fig. 2D,D') or *Abd-B* alone (Fig. 2F,F') did not affect adipocyte size or cellular homogeneity compared to the control (Fig. 2B,B'). These results suggest that ectopic expression of Abd-A or Abd-B in thoracic adipocytes amplifies Wnt signaling-induced adipocyte defects, indicating they act as key modulators of Wnt signaling effects in larval adipocytes.

If Wnt signaling-induced adipocyte heterogeneity in the abdominal region depends on Abd-A or Abd-B, we hypothesized that co-depletion of Axn and Abd-A (or Abd-B) would alleviate the adipocyte defects caused by elevated Wnt signaling in this region. Indeed, co-depletion of *Axn* and *abd-A* resulted in more uniform abdominal adipocytes compared to *Axn* depletion alone (Fig. 2M vs. Fig. 2I). Knocking down *Abd-B* also reduced adipocyte heterogeneity (Fig. 2K vs. Fig. 2I), though less effectively than *abd-A* depletion (Fig. 2K vs. Fig. 2M). Simultaneous depletion of *abd-A*, *Abd-B*, and *Axn* robustly suppressed adipocyte heterogeneity (Fig. 2O vs. Fig. 2I, quantified in Fig. 2P and Appendix

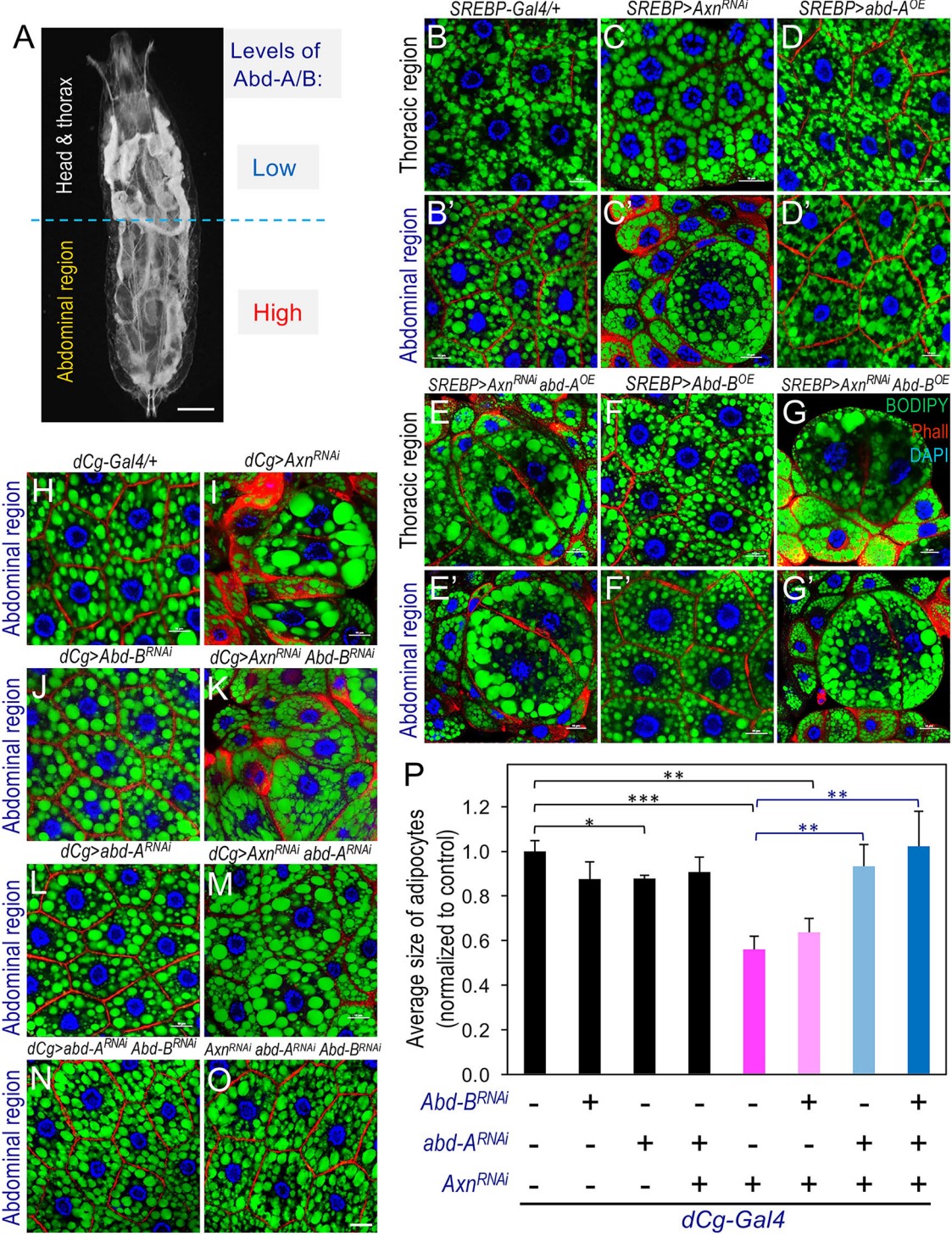

Fig. (continued).

Fig. S10A), suggesting partial redundancy between *abd-A* and *Abd-B*. In controls, individual or combined *abd-A* and *Abd-B* depletion had no obvious effect (Fig. 2L,J,N). Additionally, the transparency observed in *Axn*-depleted larvae was partially rescued by co-depletion of *abd-A* and markedly rescued when both *abd-A* and *Abd-B* were depleted (Appendix Fig. S12). These observations support the crucial roles of Abd-A and Abd-B in modulating Wnt signaling-induced fat body defects and adipocyte heterogeneity.

To further validate the interplay between Abd-A/Abd-B and Wnt signaling, we simultaneously depleted *abd-A* and *Abd-B* in the larval fat body. Depletion of *abd-A* in the context of active Wnt signaling (Appendix Fig. S8F) significantly reversed the fat body defects observed with *Axn* depletion alone (Appendix Fig. S8E). This rescue effect was even more pronounced when both *abd-A* and *Abd-B* were co-depleted (Appendix Fig. S8G vs. Appendix Fig. S8E). Additionally, we observed that Wnt-derived low TG accumulation

◀ **Figure 2. Differential expression of *abd-A* and *Abd-B* determines region-specific activation of Wnt signaling in larval adipose tissue.**

(A) Schematic diagram illustrating the transparency observed in the abdominal region of larval fat body upon activation of Wnt signaling and the relative abundance of Abd-A and Abd-B in larval adipose tissue. Scale bar: 0.5 mm. (B–G) Representative confocal images of larval adipocytes from the thoracic region (B–G) and the abdominal region (B′–G′) of the larval fat body. Genotypes are as follows: (B– B′) *+/+; SREBP-Gal4/+;* (C–C′) *UAS-Axn^RNAi^/+; SREBP-Gal4/+;* (D–D′) *UAS-Abd-A^+^/ + ; SREBP-Gal4/+* (OE: overexpression); (E–E′) *UAS-Abd-A^+^/UAS-Axn^RNAi^; SREBP-Gal4/+;* (F–F′) *UAS-Abd-B^+^/ + ; SREBP-Gal4/+;* and (G–G′) *UAS-Abd-B^+^/UAS-Axn^RNAi^; SREBP-Gal4/+.* (H–O) Confocal images of abdominal adipocytes showing significant rescue of adipocyte defects upon depleting *abd-A, Abd-B,* or both. Genotypes are as follows: (H) *dCg-Gal4/+; +;* (I) *dCg-Gal4/UAS-Axn^RNAi^; +;* (J) *dCg-Gal4/+; UAS-Abd-B^RNAi^/+;* (K) *dCg-Gal4/UAS-Axn^RNAi^; UAS-Abd-B^RNAi^/+;* (L) *dCg-Gal4/+; UAS-Abd-A^RNAi^/+;* (M) *dCg-Gal4/UAS-Axn^RNAi^; UAS-Abd-A^RNAi^/+;* (N) *dCg-Gal4/+; UAS-Abd-A^RNAi^/UAS-Abd-B^RNAi^;* and (O) *dCg-Gal4/UAS-Axn^RNAi^; UAS-Abd-A^RNAi^/UAS-Abd-B^RNAi^.* (P) Quantification of adipocyte sizes shown in (H–O) ($n = 3$ independent biological replicates, and 3–10 cells were measured in each repeat). Scale bar in (O) applies to all images from (B–B′) to (O) in this figure: 10 μm. Data are presented as mean ± SD (comparing with *dCg/+*: *$P = 0.0137$; **$P = 1.438E-03$; ***$P = 6.179E-04$; comparing with *dCg/Axn^RNAi^* and [*dCg/Axn^RNAi^+Abd-A^RNAi^*]: **$P = 4.903E-03$; *dCg/Axn^RNAi^* and [*dCg/Axn^RNAi^+Abd-A^RNAi^/Abd-B^RNAi^*]: **$P = 8.817E-03$; based on one-tailed unpaired *t* tests because our experimental model is based on well-defined directional predictions regarding Wnt signaling outcomes). Source data are available online for this figure.

levels are partially rescued upon concomitant depletion of *abd-A* and *Abd-B* (Appendix Fig. S10B). In contrast, depletions of *abd-A* alone (Appendix Fig. S8C) or combined *abd-A/Abd-B* depletion without Wnt activation (Appendix Fig. S8D) had no detectable effect on adipocyte heterogeneity. Taken together, these observations establish a causal relationship between BX-C proteins and Wnt signaling in regulating lipid homeostasis in larval adipocytes.

### Effects of Abd-A and Abd-B depletion on lipid metabolism-related gene expression in larval adipocytes

To systematically identify the target genes of Abd-A and Abd-B in larval adipocytes, we performed RNA-Seq analyses on dissected larval fat bodies with both proteins depleted (genotype: "*dCg-Gal4/ pNP-abdA-AbdB^RNAi^;+*"; abbreviated as '*AiBi*'). Depletion efficiencies were further validated through RT-qPCR analysis (Appendix Fig. S11A,B). The analysis identified 1,734 significantly upregulated genes and 1249 downregulated genes. As expected, both *abd-A* and *Abd-B* were downregulated, while other Hox genes like *Ubx* and *Antp* remained unaffected (Fig. 3B).

Pathway and Gene Ontology (GO) cluster analyses revealed signaificantly downregulated lipid metabolism-related pathways in the *AiBi* group, including 'fatty acid metabolism' (Appendix Fig. S13A), 'fatty acid degradation' (Fig. 3C), and 'fatty acid biosynthesis' (Fig. 3D). Carbohydrate metabolism-related pathways, such as 'glycolysis/gluconeogenesis', 'citrate cycle (TCA cycle)', and the 'pentose phosphate pathway', were also downregulated (Fig. 3A). In addition, genes involved in lipid particle organization (Fig. 3E), peroxisomal metabolism (Fig. 3F), autophagy (Fig. 3G), and mitophagy (Appendix Fig. S13B) were significantly affected. Notably, the depletion of *abd-A/Abd-B* reduced the expression of autophagy-related genes (Fig. 3G). This finding contrasts with previous reports that Hox proteins repress autophagy during the L3 wandering stage (Banreti et al, 2014; Duffraisse et al, 2020), potentially due to differences in larval staging. Interestingly, the co-depletion of *abd-A* and *Abd-B* also upregulated several signaling pathways, including 'Toll and Imd signaling' (Fig. 3H), 'phosphatidylinositol signaling' (Appendix Fig. S13C), 'Wnt signaling' (Appendix Fig. S13D), 'FoxO signaling' (Appendix Fig. S13E), and 'Hippo signaling' (Appendix Fig. S13F). Moreover, genes involved in 'imaginal disc-derived wing morphogenesis' were also significantly affected (Appendix Fig. S13G). These findings reveal the multifaceted roles of *BX-C* genes in regulating lipid and carbohydrate metabolism, as well as developmental signaling pathways, in larval adipocytes.

### Abd-A and Abd-B modulate the expression of Wnt-activated target genes in larval adipocytes

Our genetic analyses revealed that co-depleting *abd-A* and *Abd-B* reversed the fat accumulation defects caused by active Wnt signaling in larval adipocytes (Fig. 2H–O). To explore the mechanism, we conducted RNA-seq analyses of the abdominal fat body from *Axn*-depleted larvae and controls (*dCg-Gal4/+;+*), as Wnt signaling-induced defects were confined to the abdominal region. Depletion efficiency of *Axn* was plotted based on RNA-Seq data (Appendix Fig. S11C). As expected, the 'Wnt signaling pathway' was upregulated, while genes involved in metabolic processes, particularly 'fatty acid metabolism' and carbohydrate metabolism-related pathways such as 'glycolysis/Gluconeogenesis,' 'citrate cycle (TCA cycle),' and 'pentose phosphate pathway', were markedly downregulated (Fig. 4A). Additionally, direct Wnt signaling target genes such as *Notum, fz3,* and *nkd* were markedly upregulated (Fig. 4B).

Since active Wnt signaling can regulate target gene transcription both positively and negatively (Liu et al, 2024), we examined the effects of Abd-A and Abd-B on Wnt signaling-regulated gene expression at the cellular level. Using the *fz3-RFP* reporter as a marker for Wnt-activated genes (Olson et al, 2011), no detectable expression was observed in the control samples (Fig. 4C,C′). In contrast, *Axn* depletion in abdominal adipocytes induced strong *fz3-RFP* expression in small adipocytes with active Wnt signaling, while adjacent large adipocytes with low Wnt activity showed no expression (Fig. 4D,D′; quantified in Fig. 4G). This pattern mirrors previous observations with *nkd*, another direct Wnt target (Liu et al, 2024).

To evaluate the role of Abd-A and Abd-B in Wnt-activated target gene expression, we co-depleted them together with *Axn*. Co-depletion effectively abolished *Axn^RNAi^*-induced *fz3-RFP* expression. Specifically, co-depleting *Axn* with *abd-A* (Fig. 4F,F′ vs. Fig. 4D,D′) or *Abd-B* (Appendix Fig. S14B–B′ vs. Fig. 4D–D′) substantially reduced *fz3-RFP* expression (quantified in Fig. 4G), restoring adipocyte heterogeneity in the abdominal fat body. In contrast, depleting *abd-A* or *Abd-B* alone had little to no effect on *fz3-RFP* expression (Fig. 4E,E′; Appendix Fig. S14A,A′; quantified in Fig. 4G). To further assess the effect of Abd-A/Abd-B depletion on Wnt signaling, we performed RT-qPCR on additional Wnt-activated target genes, including *CycD, nkd,* and *Notum.* As shown in Fig. 4H, co-depletion of Abd-A and Abd-B abolished the upregulation of these genes induced by *Axn* depletion. These

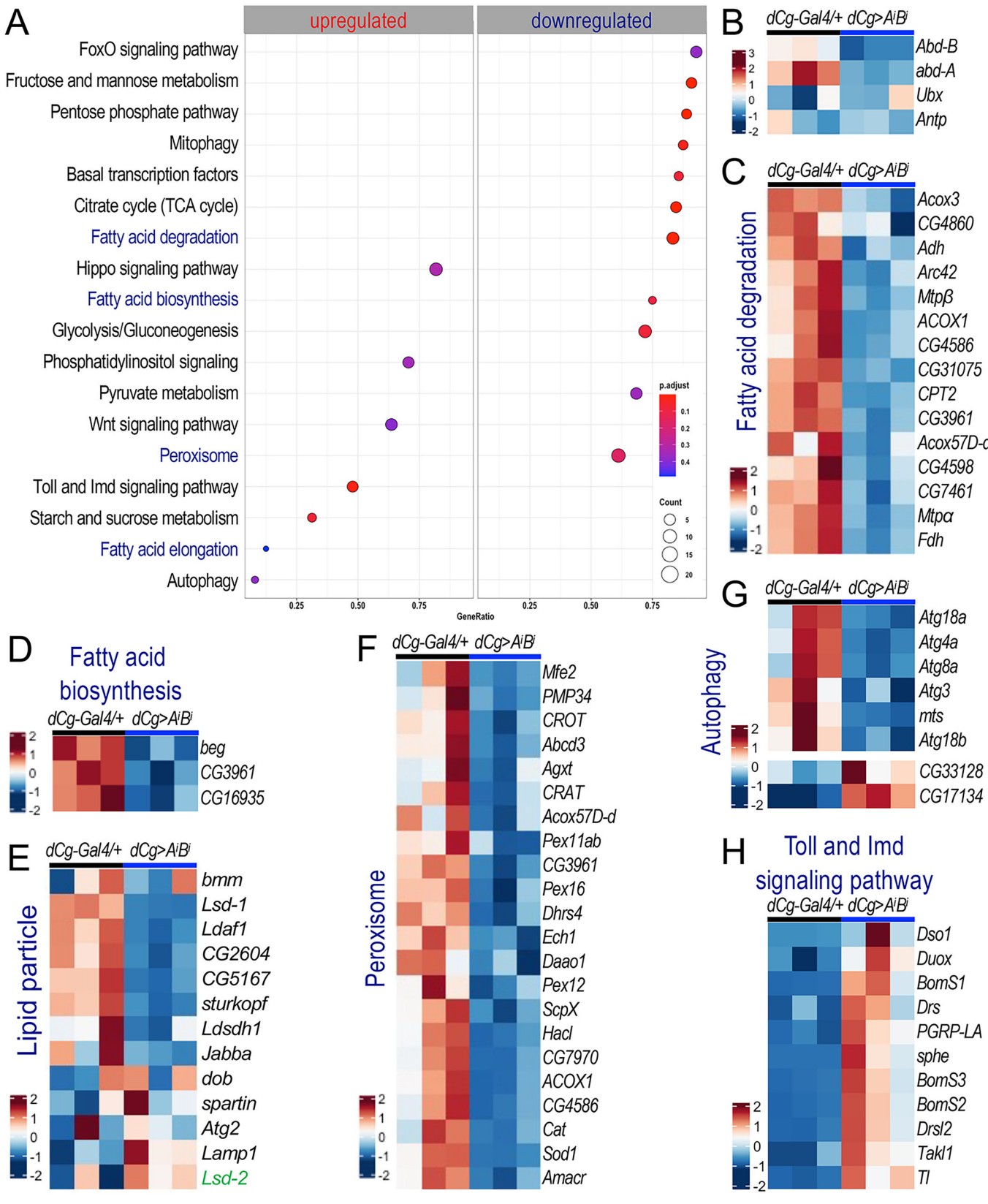

**Figure 3.   Effects of *abd-A* and *Abd-B* depletion in larval adipocytes on lipid metabolism-related gene expression.**

(A) Common pathways related to developmental signaling, lipid metabolism, and carbohydrate metabolism that are significantly altered in the larval fat body following the depletion of *abd-A* and *Abd-B* (p-value based on one-tailed unpaired *t* tests). (B) Heatmap showing the mRNA levels of *Abd-B*, *abd-A*, *Ubx*, and *Antp* in '*dCg-Gal4/+;+*' (*dCg/+*) and '*dCg-Gal4/pNP-[abd-A Abd-B]RNAi/ +*' (abbreviated as '*dCg/AiBi*'). (C–H) Heatmaps illustrating gene expression levels in various metabolic pathways: (C) 'Fatty acid degradation' pathway, (D) 'Fatty acid biosynthesis' pathway, (E) 'Lipid particle', (F) 'Peroxisome' pathway, (G) 'Autophagy,' and (H) 'Toll and Imd signaling pathway' (n = 3, independent biological repeats).

observations suggest that Abd-A and Abd-B play a permissive role in the expression of Wnt target genes in abdominal adipocytes.

## *abd-A* and *Abd-B* depletion stimulate Wnt signaling-repressed target gene expression in larval adipocytes

We next asked whether Abd-A and Abd-B also regulate the expression of Wnt-repressed genes. Pathway analyses revealed that many lipid and carbohydrate metabolism-related pathways were downregulated by active Wnt signaling (Fig. 4A). Specifically, key regulators of fatty acid biosynthesis, including *FASN1* (encoding Fatty acid synthase 1), *bgm* (*bubblegum*, encoding a long-chain-fatty-acid-CoA ligase) (Fig. 5A), as well as fatty acid degradation genes like *CPT2* (encoding carnitine palmitoyltransferase 2) and *Acox3* (Acyl-CoA oxidase 3), showed marked reductions (Fig. 5B). Additionally, lipid droplet-associated proteins (LDAPs), including *Lsd1* (encoding Lipid storage droplet-1) and *Lsd2* (encoding Lipid storage droplet-2), were also downregulated by active Wnt signaling (Fig. 5C), consistent with our previous findings (Liu et al, 2024).

To further analyze the effect of *abd-A* and *Abd-B* depletion on Wnt-repressed genes, we performed HCR RNA-FISH analysis using probes targeting *FASN1* and *Lsd2*. Consistent with our RNA-Seq data (Fig. 5A,C), depletion of *Axn*, which activates Wnt signaling, caused a marked reduction in *FASN1* and *Lsd2* expression (Fig. 5E,E' vs. Fig. 5D,D'). In contrast, depleting *abd-A* and *Abd-B* significantly increased the expression of both genes compared to controls (Fig. 5F,F' vs. Fig. 5D,D'; quantified in Fig. 5H). Moreover, co-depleting *abd-A* and *Abd-B* in a Wnt-activated background rescued the reduced expression of these genes (Fig. 5G,G' vs. Fig. 5E,E'; quantified in Fig. 5H), suggesting that Abd-A and Abd-B are required for Wnt-repressed gene expression. Consistent with the results from the *fz3-RFP* reporter (Fig. 4C–F; Appendix Fig. S14), co-depleting *abd-A* and *Abd-B* also reversed the Wnt-induced increase in *fz3* expression (Appendix Fig. S15; quantified in Fig. 5H).

Taken together, these observations support a permissive role for Abd-A and Abd-B in enabling Wnt-modulated transcriptional responses in abdominal adipocytes, affecting both Wnt-activated and Wnt-repressed target genes. This dual regulatory role is further explored below.

## Wnt signaling potentiates the expression of *abd-A* and *Abd-B* in larval adipocytes

Similar to other key genes controlling complex developmental processes, the *abd-A* and *Abd-B* loci contain large intronic regions that may integrate complex developmental cues and signaling inputs. Our CUT&RUN (Cleavage Under Targets & Release Using Nuclease) analysis using *dTCF/Pan^EGFP* wing imaginal discs (Liu et al, 2024) revealed multiple dTCF/Pan binding peaks within the

gene bodies of the *abd-A* (Fig. 6A) and *Abd-B* loci (Appendix Fig. S16C), as well as the intergenic region between them (Appendix Fig. S16B). We also performed CUT&RUN on purified nuclei from *dTCF/Pan^EGFP* larval fat bodies. Although these datasets exhibit high noise-to-signal ratios, likely due to the lipid-rich nature of adipocytes and suboptimal experimental conditions, they are presented here for comparison. As shown in Appendix Fig. S16, several dTCF/Pan-binding sites in larval adipocytes overlap with those identified in wing discs (indicated by arrows), despite the elevated background. These findings are consistent with direct regulation of *abd-A* and *Abd-B* transcription by the Wnt signaling pathway.

To test whether Wnt signaling regulates *abd-A* and *Abd-B* transcription, we used multiplexed HCR RNA-FISH to analyze their mRNA transcripts. In control larvae, *abd-A* and *Abd-B* expression was higher in abdominal adipocytes (Fig. 6C–C") and low or undetectable in thoracic adipocytes (Fig. 6B–B"). Notably, smaller adipocytes displayed higher Wnt activity when Wnt signaling is activated (Fig. 4D–D') (Liu et al, 2024). We observed that Wnt signaling activation significantly increased *abd-A* expression in abdominal adipocytes, particularly in cells with higher Wnt activity (Fig. 6E,E'), while thoracic adipocytes showed little change (Fig. 6D'). Conversely, suppressing Wnt signaling by depleting *Arrow* (*arr*) (Fig. 6F,F") or *dTCF/pan* (Fig. 6G,G") significantly reduced *Abd-A* and *Abd-B* expression in abdominal adipocytes. Together, these observations suggest that Wnt signaling might have a direct role in regulating *abd-A* and *Abd-B* transcription, enhancing their transcription under active Wnt signaling and reducing it when Wnt signaling is downregulated.

To analyze the effect of Wnt signaling on Abd-A and Abd-B protein levels, we activated Wnt signaling in larvae with Abd-A or Abd-B tagged with EGFP at their endogenous loci. In control larvae, Abd-A^EGFP levels were significantly higher in abdominal adipocytes (Fig. 6H,H') compared to the thoracic region (Appendix Fig. S17A,A'), with noticeable heterogeneity. Active Wnt signaling increased Abd-A^EGFP levels in abdominal adipocytes, particularly in the smaller, Wnt-active adipocytes (Fig. 6I,I'), whereas thoracic adipocytes showed no apparent change (Appendix Fig. S17B,B'). Similar to thoracic adipocytes, larger abdominal adipocytes lacked detectable Abd-A^EGFP expression (Fig. 6I'). These observations are noteworthy, as they further support a permissive role for Abd-A in facilitating Wnt activation in larval adipocytes, regardless of their thoracic or abdominal location.

In contrast, endogenous Abd-B^EGFP expression was low and undetectable in most larval adipocytes of the *dCg-Gal4* hetero-zygous control (Fig. 6J,J'; thoracic region shown in Appendix Fig. S17C,C'). However, active Wnt signaling increased Abd-B^EGFP expression in small abdominal adipocytes (Fig. 6K,K'; thoracic region shown in Appendix Fig. S17D,D'). Similar results were observed in ectopically tagged Abd-A^EGFP and Abd-B^EGFP lines,

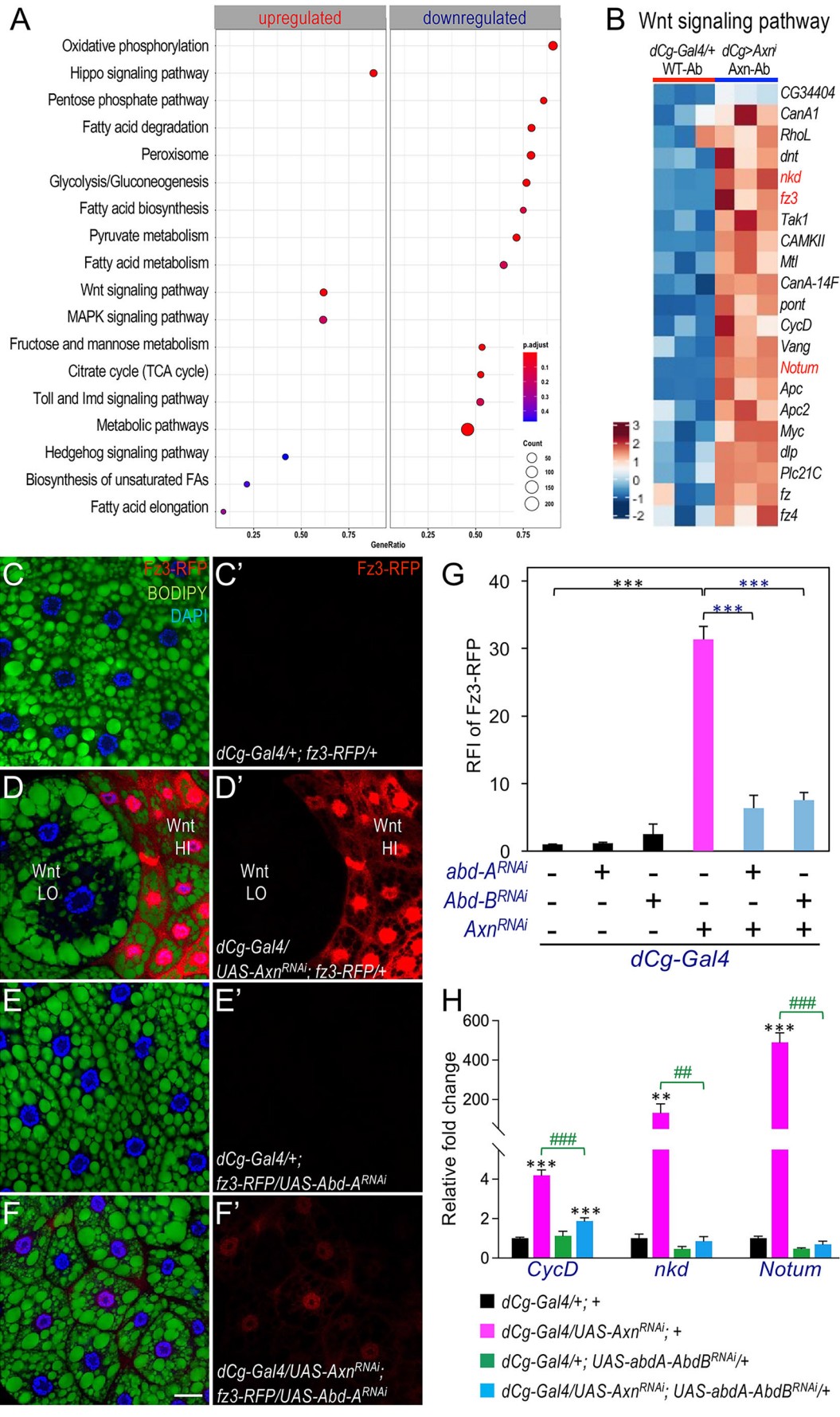

**Figure 4.  Abd-A and Abd-B are required for the expression of Wnt-activated target genes in larval adipocytes.**

(A) Common pathways related to developmental signaling, lipid metabolism, and carbohydrate metabolism that are significantly altered in the abdominal region of larval fat body upon *Axn* depletion (*dCg-Gal4/Axn^RNAi^; +*) compared to the control (*dCg-Gal4/+;+*). Gene Set Enrichment Analysis (GSEA) was carried out using permutation testing implemented in the *clusterProfiler* R package, with p-values adjusted for multiple comparisons using the Benjamini–Hochberg false discovery rate (FDR) method. (B) Heatmap representing the expression levels of genes in the 'Wnt signaling pathway' in samples from 'WT-Ab' ('Wild-type Abdominal' region of '*dCg-Gal4/+;+*' larvae) versus 'Axn-Ab' ('Axn Abdominal' region of '*dCg-Gal4/Axn^RNAi^;+*' larvae). *fz3* is highlighted in red as it was used in subsequent analysis. (C–C'–H–H') Representative confocal images of abdominal adipocytes stained with BODIPY (green) and DAPI (blue), showing *dCg-Gal4* driven depletion of *abd-A* and/or *Abd-B* along with *Axn* depletion in the *fz3-RFP* background. Genotypes are as follows: (C–C') *dCg-Gal4/+; fz3-RFP/+;* (D–D') *dCg-Gal4/UAS-Axn^RNAi^; fz3-RFP/+;* (E–E') *dCg-Gal4/+; fz3-RFP/UAS-Abd-A^RNAi^;* and (F–F') *dCg-Gal4/UAS-Axn^RNAi^; fz3-RFP/UAS-Abd-A^RNAi^*. (G) Quantification of relative fluorescence intensity (RFI) of the *fz3-RFP* reporter in a Wnt-activated background with or without *abd-A* or *Abd-B* depletion. Signal intensity in the red channel (Fz3-RFP) was measured in each image across three independent biological replicates. Data are presented as mean ± SD. Corresponding genotypes are indicated below the bar chart (comparing with *dCg/fz3* and [*dCg/fz3+Axn^RNAi^*]: ***P = 1.192E-05; comparing with [*dCg/fz3+Axn^RNAi^*] and [*dCg/fz3+Axn^RNAi^/abd-A^RNAi^*]: ***P = 3.704E-05; comparing with [*dCg/fz3+Axn^RNAi^*] and [*dCg/fz3+Axn^RNAi^/Abd-B^RNAi^*]: ***P = 5.616E-05; based on one-tailed unpaired *t* tests because our experimental model is based on well-defined directional predictions regarding Wnt signaling outcomes). (H) RT-qPCR analysis of Wnt-activated target genes *CycD*, *nkd*, and *Notum* (n = 3, independent biological repeats). The corresponding genotypes are color-coded and indicated below the bar chart. Scale bar in (F) applies to all images from (C–C') to (F–F') in this figure: 10 µm. Asterisks (*) indicate comparisons with the '*dCg-Gal4/+*' control, while pound signs (#) indicate comparisons with '*dCg-Gal4/Axn^RNAi^; +*'. Data are presented as mean ± SD (For *CycD*: comparing with *dCg/+* and [*dCg/Axn^RNAi^*] ***P = 4.021E-05; comparing with *dCg/+* and [*dCg/abd-A^RNAi^,Abd-B^RNAi^*] ***P = 9.522E-04; ###P = 2.462E-04. For *nkd*: **P = 7.228E-03; ##P = 7.202E-03. For *Notum*: ***P = 7.141E-05; ###P = 8.026E-04; based on one-tailed unpaired *t* tests because our experimental model is based on well-defined directional predictions regarding Wnt signaling outcomes). Source data are available online for this figure.

generated using re-engineered BAC clones to introduce EGFP tags (Kudron et al, 2018) (Appendix Fig. S17E,E' vs. Appendix Fig. S17F,F', and Appendix Fig. S17G,G' vs. Appendix Fig. S17H,H'). Taken together, these observations suggest that active Wnt signaling directly stimulates *abd-A* transcription, and to a lesser extent, *Abd-B*, in abdominal adipocytes.

## Wnt targets are predominantly expressed in the abdominal region of the larval fat body

Our results presented above suggest that both Abd-A and Abd-B positively modulate Wnt signaling (Fig. 2), which in turn directly stimulates their own transcription (Fig. 6). This interaction reveals a crosstalk between the Abd-A/Abd-B and Wnt signaling pathways (Fig. 7A). Given the significantly higher expression of *abd-A* and *Abd-B* in abdominal compared to thoracic adipocytes (Fig. 1J), we further analyzed how this interplay contributes to regional differences among larval adipocytes. To address this, we performed RNA-seq on dissected thoracic and abdominal fat bodies from *Axn*-depleted larvae. Control thoracic fat body samples were designated as 'WT-Th' (wild-type thoracic), abdominal samples as 'WT-Ab' (wild-type abdominal), *Axn^RNAi^* thoracic fat bodies as 'Axn-Th' (*Axn^RNAi^* thoracic), and *Axn^RNAi^* abdominal samples as 'Axn-Ab' (*Axn^RNAi^* abdominal) (Fig. 7B). RNA-seq analyses confirmed that *Axn* was depleted in both 'Axn-Th' and 'Axn-Ab' compared to controls (Fig. 7C). As expected, *abd-A* and *Abd-B* mRNA levels were higher in abdominal adipocytes ('WT-Ab') than in thoracic adipocytes ('WT-Th'; Fig. 7C). Activation of Wnt signaling via *Axn* depletion further increased *abd-A* expression in abdominal adipocytes ('Axn-Ab') compared to thoracic adipocytes ('Axn-Th'), while the effect on *Abd-B* expression level was marginal (Fig. 7C).

Our analysis of candidate genes, such as *fz3*, *CycD*, *nkd*, and *Notum* (Fig. 4), as well as *FASN1* and *Lsd-2* (Fig. 5), suggested a permissive role of Abd-A and Abd-B in modulating both Wnt-activated and Wnt-repressed gene expression. Using DAVID GO analysis of our RNA-seq data, we identified a set of genes categorized under the 'Wnt signaling pathway,' including known Wnt-targets like *fz3*, *nkd*, *Notum*, and *CycD* (Fig. 7D). As expected,

these Wnt target genes showed low expression in control fat bodies (both 'WT-Th' and 'WT-Ab'). However, Wnt signaling activation in Axn-Ab significantly upregulated these genes compared to Axn-Th, where their expression remained low and resembled that of the control samples ('WT-Th' and 'WT-Ab') (Fig. 7D). Interestingly, similar trends were observed for genes associated with other key signaling pathways, such as 'MAPK signaling' (Appendix Fig. S18A), 'Hedgehog signaling' (Appendix Fig. S18B), and 'Hippo signaling' (Appendix Fig. S18C). Gene expression was predominantly enhanced in 'Axn-Ab' compared to 'Axn-Th', which closely resembled the control regions ('WT-Th' and 'WT-Ab'). These results further support the role of Abd-A and Abd-B in facilitating Wnt-activated gene expression in abdominal adipocytes (Fig. 7A).

We also identified a set of Wnt signaling-repressed genes involved in regulating both lipogenesis and lipid mobilization (Liu et al, 2024), categorized under the 'fatty acid metabolism' pathway. In control samples, these genes were expressed at similar levels in thoracic ('WT-Th') and abdominal ('WT-Ab') adipocytes (Fig. 7E). However, their expression levels were significantly reduced in Wnt-active abdominal adipocytes ('Axn-Ab') compared to Wnt-active thoracic adipocytes ('Axn-Th'). This trend was consistent across genes associated with 'fatty acid biosynthesis' (Appendix Fig. S18D), 'fatty acid degradation' (Appendix Fig. S18E), 'Toll and Imd signaling pathway' (Appendix Fig. S18F), and peroxisome-related genes (Fig. 7F). Notably, the suppression of peroxisomal genes by Wnt signaling was primarily restricted to the abdominal region ('Axn-Ab') (Fig. 7F). These findings suggest that Abd-A and Abd-B are required not only for activating Wnt target genes but also for repressing genes involved in lipid metabolism and developmental signaling pathways in larval adipocytes.

Collectively, our results suggest that Abd-A and Abd-B play a permissive role in regulating both Wnt-activated and Wnt-repressed gene expression in larval adipocytes, with their effects being most pronounced in abdominal adipocytes where endogenous *abd-A* and *Abd-B* are expressed (Fig. 7A). The presence of Abd-A and Abd-B in abdominal adipocytes enables them to respond to Wnt signaling by promoting lipid mobilization and repressing lipogenesis. In contrast, adipocytes lacking *abd-A* and *Abd-B* expression, whether in the thoracic or abdominal regions, fail to

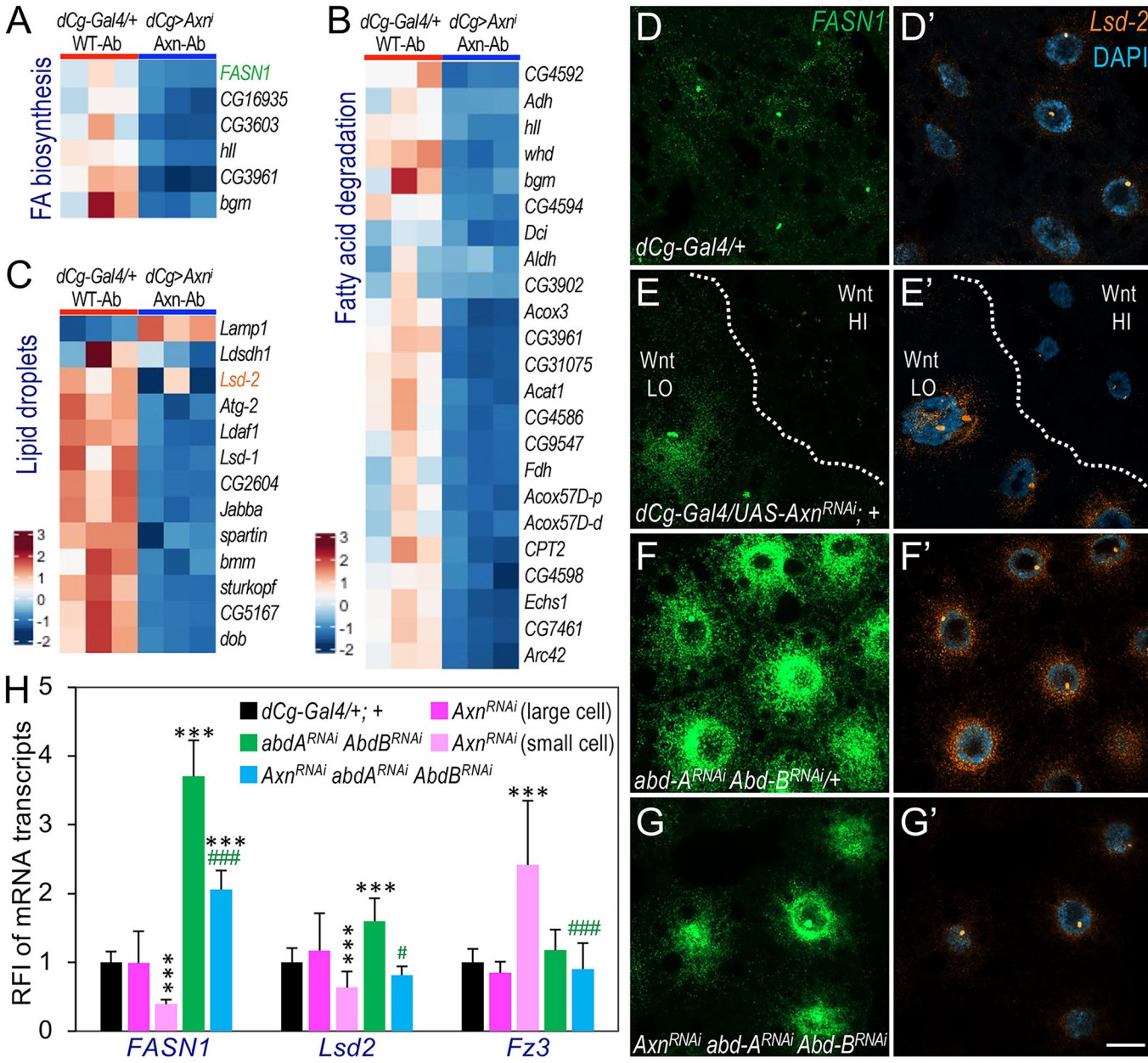

**Figure 5. Abd-A and Abd-B are required for the expression of Wnt-repressed target genes in larval adipocytes.**

(A–C) Heatmaps showing the gene expression levels of the 'Fatty acid biosynthesis' (A), 'Fatty acid degradation' (B), and 'Lipid particle' (C) pathways in WT-Ab ('Wild-type Abdominal' region of 'dCg-Gal4/+;+' larvae) vs. Axn-Ab ('Axn Abdominal' region of 'dCg-Gal4/Axn$^{RNAi}$;+' larvae). FASN1 in (A) and Lsd-2 in (C) are highlighted in green and orange, respectively, as these genes were used in the subsequent analyses. (D–D'–G–G') Representative confocal images showing mRNA transcripts of FASN1 (green) and Lsd-2 (orange) in abdominal adipocytes, detected by the HCR RNA-FISH assay. Large adipocytes, marked as "Wnt LO" to indicate low Wnt/Wg signaling activity, are outlined with dotted lines in the left half of the images (E–E'), while adjacent smaller adipocytes, labeled "Wnt HI," represent cells with high Wnt/Wg signaling activity in the right half of the images. Genotypes: (D–D') dCg-Gal4/+, (E-E') dCg-Gal4/Axn$^{RNAi}$; +, (F–F') dCg-Gal4/+; abd-A$^{RNAi}$/ Abd-B$^{RNAi}$, and (G–G') dCg-Gal4/Axn$^{RNAi}$; abd-A$^{RNAi}$/ Abd-B$^{RNAi}$. Scale bar in (G') applies to all images from (D–D') to (G–G'): 20 μm. (H) Quantification of FASN1, Lsd-2, and fz3 mRNA transcripts. For each genotype, the fluorescent intensities of four to 23 cells were measured using ImageJ, and the relative fluorescent intensity (RFI) was normalized to the control genotype (dCg-Gal4/+). Asterisks (*) indicate comparisons with the 'dCg-Gal4/+' control, while pound signs (#) indicate comparisons with 'dCg-Gal4/Axn$^{RNAi}$; +' (small cells). Data are presented as mean ± SD (For FASN1: comparing with dCg/+ and [dCg/Axn$^{RNAi}$] ***P = 6.860E-11; comparing with dCg/+ and [dCg/abd-A$^{RNAi}$,Abd-B$^{RNAi}$] ***P = 5.557E-13; ###P = 2.242E-12. For Lsd-2: comparing with dCg/+ and [dCg/Axn$^{RNAi}$] ***P = 5.537E-04; comparing with dCg/+ and [dCg/abd-A$^{RNAi}$,Abd-B$^{RNAi}$] ***P = 5.879E-05; #P = 0.0448. For fz3: ***P = 1.086E-06; ###P = 2.294E-08; based on one-tailed unpaired t tests because our experimental model is based on well-defined directional predictions regarding Wnt signaling outcomes). Source data are available online for this figure.

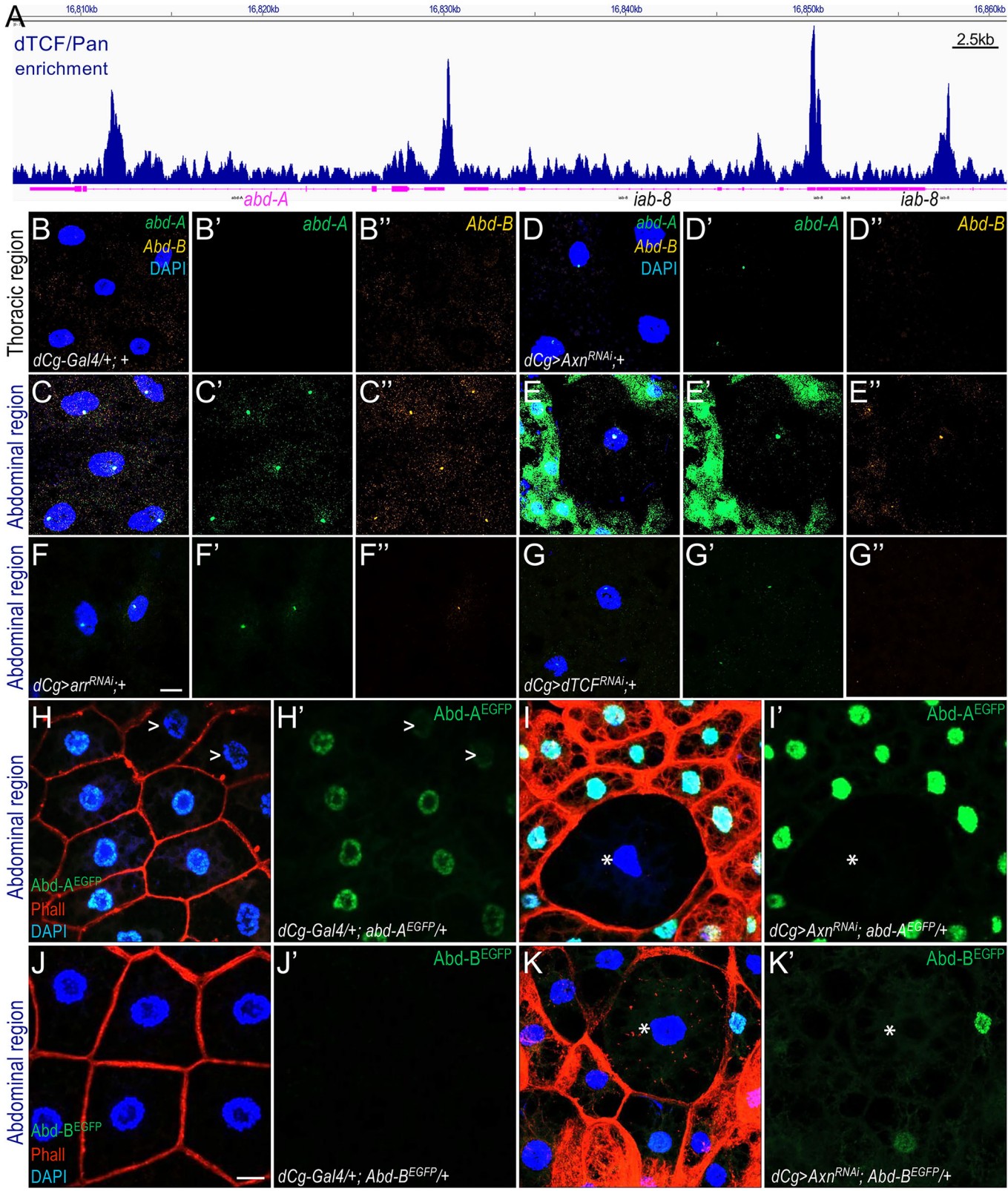

**Figure 6. Wnt signaling may directly regulate the expression of *abd-A* and *Abd-B* in larval adipocytes.**

(A) Screenshot from the Integrative Genomics Viewer (IGV) browser showing dTCF/Pan binding peaks at the *abd-A* and part of the *iab-8* loci, detected via CUT&RUN assay performed on wing discs. The y-axis is autoscaled, and the different isoforms of these genes are collapsed and displayed in magenta. Scale bar: 2.5 kb. (B–B″–G–G″) Representative confocal images showing the mRNA transcripts of *abd-A* (green) and *Abd-B* (orange) in thoracic (B–B″, D–D″) and abdominal (C–C″, E–E″, F–F″, G–G″) adipocytes, detected by HCR RNA-FISH. These images were acquired and processed using identical confocal microscopy settings. Genotypes: (B–B″, C–C″) *dCg-Gal4/+;* +, (D–D″, E–E″) *dCg-Gal4/UAS-Axn^RNAi;* +, (F–F″) *dCg-Gal4/UAS-Arr^RNAi;* +, and (G–G″) *dCg-Gal4/UAS-dTCF^RNAi;* +. Scale bar in (F) applies to all the images shown in (B–B″) to (G–G″): 10 μm. (H, I) Depleting *Axn* in the Abd-A^EGFP background. Genotypes: (H–H′) *dCg-Gal4/+; Abd-A^EGFP/+* (control) and (I–I′) *dCg-Gal4/UAS-Axn^RNAi; Abd-A^EGFP/+*. Adipocytes with lower levels of Abd-A^EGFP expression are marked with '>', and adipocytes with low Wnt activity are marked with an asterisk (*). (J–K′) Depletion of *Axn* in the Abd-B^EGFP background. Genotypes: (J–J′) *dCg-Gal4/+; Abd-B^EGFP/+* (control) and (K–K′) *dCg-Gal4/UAS-Axn^RNAi; Abd-B^EGFP/+*. Adipocytes with low Wnt activity are marked with an asterisk (*). Scale bar in (J) applies to images (H–H′) to (K–K′): 10 μm. Source data are available online for this figure.

activate or repress Wnt target genes, leaving thoracic and occasional abdominal adipocytes unresponsive to Wnt signaling-induced lipid mobilization.

## Discussion

This study reveals that the differential expression of BX-C proteins Abd-A and Abd-B defines adipocyte heterogeneity in *Drosophila* larvae, a process further amplified by active Wnt/Wg signaling. Abd-A and Abd-B exert a permissive effect on Wnt/Wg target gene transcription, while Wnt/Wg signaling, in turn, enhances *abd-A* and *Abd-B* expression specifically in abdominal adipocytes (Fig. 7A). This reciprocal regulatory interplay reinforces larval adipocyte heterogeneity and fine-tunes lipid homeostasis in *Drosophila*.

### The role of Abd-A and Abd-B in regulating lipid metabolism

Previous studies have shown the expression of *BX-C* proteins in the abdominal fat body of *Drosophila* larvae using immunostaining techniques (Marchetti et al, 2003). However, the functional significance of these proteins in lipid metabolism has remained unexplored. Our transcriptomic analyses of '*AiBi*' adipocytes showed a marked downregulation of metabolic pathways, including those involved in carbohydrate and fatty acid metabolism. Further analyses revealed the effect of *abd-A* and *Abd-B* depletion on various metabolic processes, such as the 'pentose phosphate pathway,' 'TCA cycle,' 'fatty acid degradation,' 'fatty acid degradation,' and 'fatty acid biosynthesis'. These findings suggest the physiological role of Abd-A and Abd-B in regulating lipid and carbohydrate metabolism in larval adipocytes.

It remains unclear whether Abd-A and Abd-B directly regulate the transcription of those lipid metabolism-related genes or act through indirect mechanisms. Addressing this question requires ChIP-seq or related analyses, though technical challenges must be overcome. The development of ChIP-grade antibodies specific to Abd-A and Abd-B, or the use of EGFP-tagged Abd-A and Abd-B strains, offers potential solutions for ChIP-seq or CUT&RUN analyses. However, the abundance of lipid droplets and the limited number of larval adipocytes present significant obstacles, as evidenced by our previous CUT&RUN attempts, which suffered from poor signal-to-noise ratios (Appendix Fig. S16) (Liu et al, 2024). Performing CUT&RUN in other tissues, such as the central nervous system (Duckhorn, 2022), or at different stages, like late embryogenesis, may provide alternative avenues for investigation but could introduce additional complexities. Overcoming these

technical hurdles will require future advancements in technology. Interestingly, in mammals, HOXB13, a homolog of Abd-B, has been shown to repress genes involved in de novo lipogenesis and steroid metabolism (Lu et al, 2022). Specifically, depletion of HOXB13 increased the expression of lipogenic genes in human prostate cancer LNCaP cells, with ChIP-qPCR analyses showing HOXB13 enrichment at the enhancers of genes such as *FASN*, suggesting that HOXB13 may directly repress their transcription (Lu et al, 2022).

The Hox family of transcription factors can function as either transcriptional activators or repressors (Afzal and Krumlauf, 2022; Bondos et al, 2020). For example, studies using the *dpp674-lacZ* reporter, driven by a visceral mesoderm-specific enhancer of *decapentaplegic* (*dpp*) during embryogenesis, show that Abd-A functions as a repressor, while Ubx and Abd-B act as activators (Capovilla and Botas, 1998). However, Abd-A can also act as an activator in certain contexts. For instance, ectopic expression of *abd-A* in male histoblast nest cells induces *wg* expression in the seventh segment of early pupal abdominal epithelia (Singh and Mishra, 2014). Additionally, in the cardiac tube, Abd-A activates the transcription of *Ih* (encoding a voltage-gated ion channel) and *ndae1* (encoding a Na⁺-driven anion exchanger) during embryogenesis but represses *Ih* expression during the pupal stage (Monier et al, 2005; Perrin et al, 2004). These observations suggest that Abd-A can toggle between functioning as a transcriptional activator or repressor, with this switch being determined by interactions involving the Hox hexapeptide motif, the PFER motif, and a poly(Q) activation domain (Merabet et al, 2003). Elucidating how Abd-A modulates lipid metabolism-related gene expression in larval adipocytes may uncover additional regulatory mechanisms underlying its dual role as both an activator and a repressor.

### The interplay between Abd-A/Abd-B and Wnt signaling

Our genetic and cell biological analyses reveal that the reduced fat accumulation caused by active Wnt signaling is mitigated by depleting *abd-A* and *Abd-B*, suggesting three possible explanations for this phenomenon. First, Abd-A and Abd-B may regulate lipid metabolism through direct interaction with dTCF/Pan (Baeza et al, 2015; Bischof et al, 2018), the key transcription factor downstream of Wnt signaling (Barker and Clevers, 2006; Molenaar et al, 1996). Second, the effect of Abd-A and Abd-B on lipid metabolism might be direct, involving shared target genes with Wnt signaling. However, identifying these direct target genes in larval adipocytes remains a technically challenging task. Third, the interplay between Abd-A/Abd-B and additional signaling pathways, such as Hippo and TGF-β signaling pathways (Massague, 2012; Varelas, 2014; Varelas et al, 2010), may also contribute. Upregulation of these pathways in the '*AiBi*'

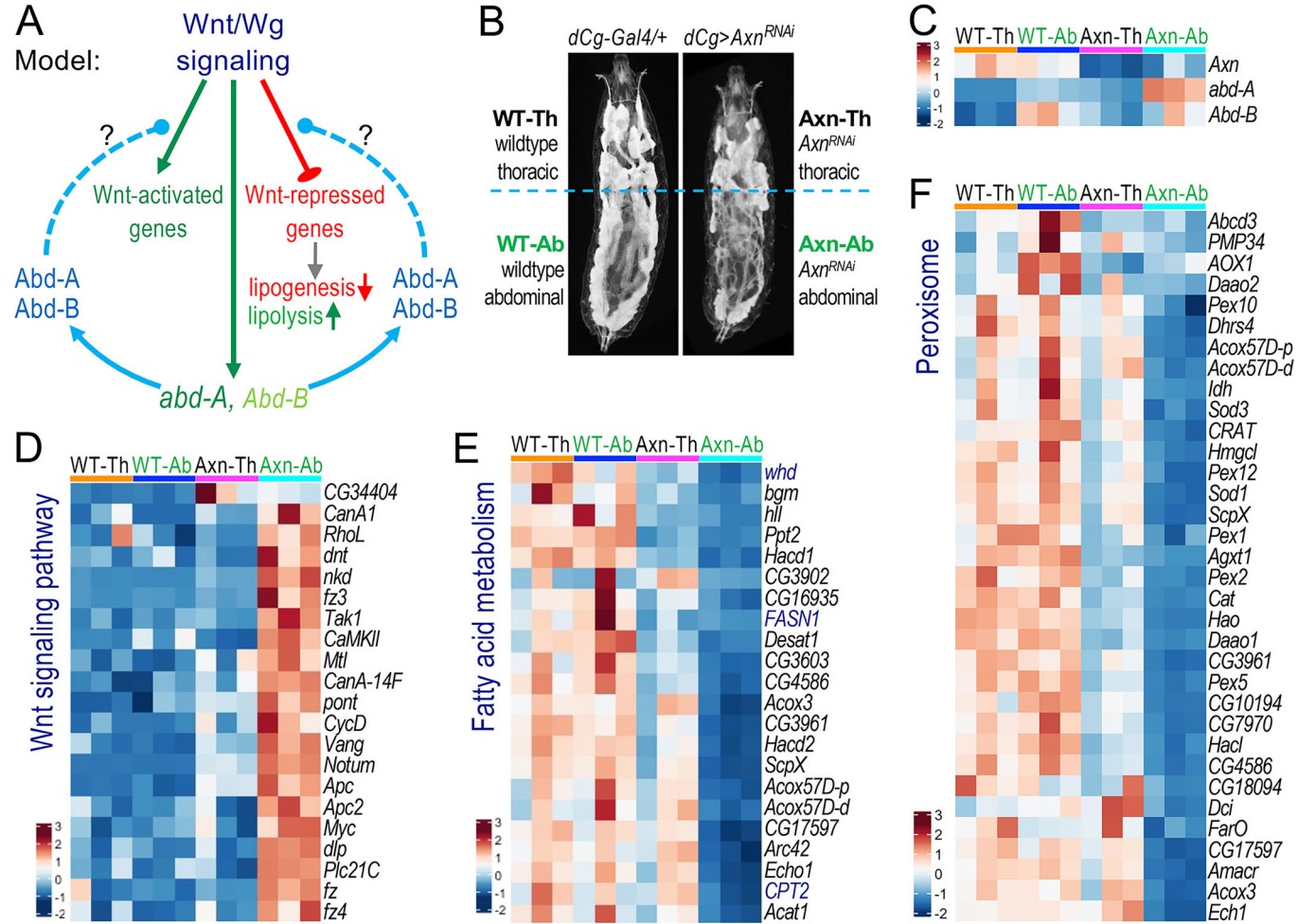

**Figure 7. Crosstalk between Wnt/Wg signaling and Abd-A/Abd-B defines regional difference in the larval fat body.**

(A) The proposed model illustrates the crosstalk between Wnt/Wg signaling and Abd-A/Abd-B. Wnt signaling might have a direct role in regulating *abd-A* expression and, to a lesser extent, *Abd-B*. Both Abd-A and Abd-B proteins are required for the activation and repression of Wnt target gene expression. We propose that this model explains the interplay between Wnt/Wg signaling and Abd-A/Abd-B in most of the abdominal adipocytes. In contrast, this genetic circuit is inactive in thoracic adipocytes and in occasional abdominal adipocytes lacking Abd-A and Abd-B expression. The exact molecular mechanism by which Abd-A and Abd-B exert their permissive role in modulating Wnt target gene expression remains unknown and is indicated by dashed lines and question marks in the model. (B) Whole larvae images showing distinct regions used for RNA-seq analysis. Thoracic and abdominal regions of control larvae (*dCg-Gal4/+;+*) are labeled as 'WT-Th' and 'WT-Ab', respectively. The thoracic and abdominal regions of '*dCg-Gal4/UAS-Axn$^{RNAi}$; +*' larvae are labeled as 'Axn-Th' and 'Axn-Ab', respectively. (C) Heatmap representing mRNA levels of *Axn*, *abd-A*, and *Abd-B* in 'WT-Th', 'WT-Ab', 'Axn-Th', and 'Axn-Ab' samples. (D) Heatmap representing gene expression levels of the 'Wnt signaling pathway,' showing prominent Wnt-activated gene expression in 'Axn-Ab,' while 'Axn-Th' resembles 'WT-Th' and 'WT-Ab'. (E) Heatmap showing the gene expression levels of 'Fatty acid metabolism' genes, and (F) heatmap showing the gene expression levels of 'Peroxisome' related genes, both demonstrating that Wnt-repressed gene expression is significantly reduced in 'Axn-Ab,' while 'AT' resembles 'WT-Th' and 'WT-Ab' (*n* = 3, independent biological repeats).

background suggests that the interactions among these signaling pathways could influence lipid metabolism in response to Wnt signaling. These scenarios are not mutually exclusive, and the complexity of these interactions warrants further investigation.

An unexpected finding is that the depletion of Abd-A and Abd-B affects both Wnt-activated and Wnt-repressed genes, suggesting a permissive role of these proteins in regulating Wnt target gene expression in larval adipocytes. While the mechanism remains unclear, we speculate that transcription cofactors essential for both the activation and repression of Wnt target genes may be regulated by Abd-A and Abd-B. Parallels from human studies support this idea. For example, depleting HOXB5 represses *β-catenin* expression and downstream Wnt target genes, thereby inhibiting tumor

invasion in non-small cell lung cancer (Yu et al, 2020). Similarly, HOXB7 interacts directly with β-catenin, and its depletion inhibits Wnt/β-catenin signaling, while HOXB8 prevents Wnt/β-catenin activation, thereby suppressing tumorigenesis and metastasis in colorectal cancer (Yu et al, 2020). As *HOXB7* and *HOXB8* are human orthologs of *Ubx* and *abd-A*, respectively (Pearson et al, 2005), these findings suggest a potentially conserved role of HOX proteins in modulating Wnt target gene transcription from flies to humans.

Another intriguing observation is the upregulation of *abd-A* and *Abd-B* transcription by active Wnt signaling (Fig. 6). Our CUT&RUN analysis identified multiple dTCF/Pan binding sites, suggesting that Wnt signaling may directly stimulate *abd-A* and

*Abd-B* transcription. However, this effect is restricted to adipocytes where *abd-A* and *Abd-B* are already expressed, predominantly in abdominal adipocytes. This indicates that Wnt signaling does not instruct the transcriptional activation of *abd-A* and *Abd-B* but synergizes with other factors to amplify their expression. Interestingly, the *HoxB9* gene, an ortholog of *Abd-B* (Pearson et al, 2005), has been shown to be a direct target of WNT/TCF signaling in mice (Nguyen et al, 2009), further suggesting a conserved regulatory relationship between Wnt signaling and *HOX* gene expression across species.

These mechanisms, coupled with the exclusive expression of Abd-A and Abd-B in the abdominal adipocytes, explain why active Wnt signaling specifically affects abdominal adipocytes while having little impact on thoracic adipocytes (Fig. 7A). Interestingly, large adipocytes are occasionally observed in the abdominal region under active Wnt signaling (Liu et al, 2024; Zhang et al, 2017). Strikingly, these large adipocytes lack Abd-A and Abd-B expression and respond to Wnt signaling in a manner similar to thoracic adipocytes. Thus, our model (Fig. 7A) provides a mechanistic explanation for this adipocyte heterogeneity within the abdominal region as well. In summary, our transcriptomic and cell biological analyses reveal the critical roles of *abd-A* and *Abd-B* in regulating lipid homeostasis and the transcription of Wnt target genes. This regulation is further potentiated by active Wnt signaling, revealing the intricate interplay between these key factors in defining adipocyte heterogeneity and lipid metabolism.

Notably, we have observed consistent reductions in lipid storage in both *Axn*$^{127}$ mutants and in fat body-specific *Axn* knockdown using three different Gal4 drivers (*dCg-Gal4*, *SREBP-Gal4*, and *r4-Gal4* lines), as reported in our previous studies (Liu et al, 2024; Zhang et al, 2017). Furthermore, similar phenotypes were observed upon *slmb* depletion (Liu et al, 2024) or overexpression of *wg* in the larval fat body using *dCg-Gal4* (Lee et al, 2014), both of which led to reduced fat accumulation, particularly in the abdominal region—closely resembling the phenotypes seen in *Axn*$^{127}$ mutant or fat body-specific *Axn* knockdown larvae. These findings collectively support a specific and robust role for Wnt/Wg signaling pathway activation in modulating lipid homeostasis in the larval fat body. Moreover, in our previous study, by generating Flp-FRT clones of *Axn* mutant alleles (*Axn*$^{127}$ and *Axn*$^{S044230}$) in the larval fat body, we observed a marked reduction in fat accumulations specifically within *Axn* mutant clones, but not in the surrounding wild-type adipocytes (Zhang et al, 2017). This observation supports a cell-autonomous role of Wnt signaling in regulating lipid metabolism. To avoid the confounding effects of temperature, we opted not to utilize Gal80$^{ts}$ system, which requires temperature shifts. Since temperature significantly influences various metabolic processes, RNAi efficiency based on the Gal4-UAS system, and developmental timing, its use would have introduced additional variables and complexity into our analyses of Wnt signaling in regulating lipid homeostasis.

## The heterogeneous expression of Abd-A and Abd-B defines adipocyte heterogeneity

The transcriptional regulation of the *BX-C* locus has been extensively studied for its crucial role in establishing the anterior-posterior segmental body plan during embryogenesis (Garcia-Fernandez, 2005; Lewis, 1978; Struhl, 1981). However,

while this regulation is thought to be maintained during postembryonic stages by *PcG* and *TrxG* proteins (Muller and Kassis, 2006; Oktaba et al, 2008; Schuettengruber et al, 2007; Schwartz and Pirrotta, 2007), direct evidence supporting their involvement in regulating *BX-C* genes in the larval fat body remains unknown.

In *Drosophila* larvae, ~2100–2500 adipocytes form a cohesive sheet along the anterior-posterior body axis (Rizki, 1978). Previous research by Rizki and colleagues identified differences among larval adipocytes based on variations in autofluorescent metabolites, such as kynurenine and pteridines (Rizki, 1978). Additional studies suggest that specific body regions may provide positional cues for the functional specialization of fat cells, contributing to the biochemical diversity of the larval and adult fat body (Miller et al, 2002). Our study builds upon these findings by demonstrating that endogenous *abd-A* and *Abd-B* are heterogeneously expressed in larval fat body, supporting the earlier reports of differential expression of these proteins (Marchetti et al, 2003). Contrary to the conventional view of the larval fat bodies as homogeneous tissues, our RNA-seq analyses of thoracic and abdominal fat bodies, along with previous studies (Marchetti et al, 2003; Rizki, 1978), reveal substantial heterogeneity. Furthermore, we show that Wnt signaling amplifies this adipocyte heterogeneity, influencing the regulation of lipid homeostasis and potentially other physiological functions. However, beyond lipid metabolism, the functional significance of adipocyte heterogeneity across different body regions warrants further investigation using more advanced analytical approaches (see below).

Our HCR analyses reveal that active Wnt signaling predominantly stimulates higher Abd-A expression in small, Wnt-active adipocytes, while large adipocytes show only moderate changes in *abd-A* expression (Fig. 6E'). Similar patterns were observed for EGFP-tagged Abd-A. Weaker but consistent effects were also noted for *Abd-B* mRNA and Abd-B$^{EGFP}$ expression (Fig. 6). Combined with our genetic data, which suggest a permissive role of Abd-A and Abd-B in regulating Wnt target gene expression, these observations imply that lower *abd-A* and *Abd-B* expression in large abdominal adipocytes and thoracic adipocytes diminishes Wnt activity and Wnt-induced lipid mobilization. Thus, the heterogeneous expression of *abd-A* and *Abd-B* plays a key role in larval adipocyte heterogeneity. Further investigation into the genetic circuits controlling this heterogeneity in larval adipocytes will advance our understanding of *BX-C* gene regulation beyond embryogenesis.

## Evolution and physiological relevance of adipocyte heterogeneity: *"Vive la différence"*

Adipocytes play a crucial role in regulating lipid homeostasis and energy metabolism in metazoans, and their dysregulation is linked to disorders such as obesity, diabetes, and cancer. In mammals, three major adipocyte types (white, brown, and beige) have been characterized (Pfeifer and Hoffmann, 2015; Shinde et al, 2021; Wang and Seale, 2016). White adipocytes primarily store triglycerides, brown adipocytes specialize in thermogenesis, and beige adipocytes share functional and cellular features of both. Despite extensive research in mice and humans, the evolutionary origins and adaptive significance of adipocyte diversification remain poorly understood. In this study, we identified a genetic

mechanism that defines larval adipocyte heterogeneity in *Drosophila*. We propose that this cellular diversity may represent an ancestral form of adipocyte specialization that emerged early in invertebrate evolution.

In addition to adipocyte types, their distribution varies by sex and developmental stage, correlating with disease susceptibilities (Tchkonia et al, 2013). For example, visceral obesity is linked to increased risk of metabolic diseases and systemic metabolic dysfunction (Tchkonia et al, 2013). Visceral adipocytes are less sensitive to insulin-mediated suppression of lipolysis compared to leg adipocytes in humans (Meek et al, 1999). Similarly, in our study, abdominal adipocytes showed heightened sensitivity to Wnt/Wg signaling-induced lipid mobilization, driven by specific expression of *abd-A* and *Abd-B*. Depleting *abd-A*, *Abd-B*, or *Axn* altered the expression of genes involved in several signaling pathways, such as Hedgehog, Hippo, MAPK, Toll, and Imd signaling pathways (Fig. 3; Appendix Fig. S18). This indicates that ligands for these pathways may be secreted by other tissues or organs into body cavities, enabling inter-organ communication. It would be interesting to explore whether adipocytes in different fat body regions exhibit distinct sensitivities to these pathways, as observed with Wnt signaling.

Studying adipocyte diversity in invertebrates provides valuable insights into evolutionary adaptations. While genes associated with lipogenesis and lipolysis exist in unicellular organisms such as *Saccharomyces cerevisiae* and *Candida parapsilosis* (Daum et al, 2007; Neugnot et al, 2002; Zweytick et al, 2000), insects are the first organisms to feature dedicated adipocytes for fat storage (Parra-Peralbo et al, 2021). This specialization distinguishes insects from lower multicellular animals, such as nematodes, where lipid droplets are stored in gut epithelial cells. Beyond fat storage, insect adipose tissue also performs endocrine and immune functions, resembling those of higher organisms (Azeez et al, 2014). We speculate that adipocyte heterogeneity in *Drosophila* may represent a primitive analogue of the white and brown adipocytes found in mammals, albeit lacking their sophisticated functions and complex regulatory networks (Hemba-Waduge et al, 2025; Parra-Peralbo et al, 2021). Notably, larval adipocytes display intrinsic differences that become more pronounced under hyperactive Wnt/Wg signaling through the genetic circuit uncovered in this work.

Although the physiological significance of distinct fat body regions and larval adipocyte populations remains unclear, studies in Lepidoptera suggest that regional differentiation within fat body correlates with the segregation of synthetic and storage functions (Haunerland and Shirk, 1995). Several speculative scenarios may further explain the physiological relevance of adipocyte heterogeneity. First, subtle differences among adipocytes may provide adaptive advantages under stress conditions such as starvation, nutrient fluctuations, temperature changes, or bacterial infection. Because all of our analyses were performed under standard conditions, further studies should examine whether regional differences in the larval fat body become more pronounced under physiological perturbations or pathological challenges. Second, it is noteworthy that thoracic adipocytes in wild-type larvae exhibit strong kynurenine autofluorescence (KAF$^+$) compared to abdominal adipocytes (Butterworth et al, 1988; Rizki, 1961). Given that tryptophan and kynurenine are precursors of the eye pigment ommochrome and that thoracic adipocytes are located near the eye imaginal discs, these adipocytes may facilitate metabolite shuttling to the discs during metamorphosis (Rizki, 1978). Third, adipocytes

in different fat body regions may serve distinct structural and developmental roles. The ventral arm of the thoracic fat body, for instance, contacts and cushions organs such as the salivary glands, central nervous system (brain and ventral nerve cord), and imaginal discs (labial, eye, and leg discs). Preferential mobilization of abdominal lipid stores could meet acute energy demands (e.g., during wandering or pupariation) while preserving lipids in regions essential for structural or developmental functions.

Regional specialization of the fat body has also been documented in other insects (Haunerland and Shirk, 1995). Our proposed model (Fig. 7A) offers a testable framework for elucidating the molecular mechanisms underlying adipocyte heterogeneity in insects and potentially in mammals. This study examined the role of BX-C proteins in larval adipocytes and their contribution to adipocyte heterogeneity in *Drosophila* larvae. Given the evolutionary conservation of both the Wnt signaling pathway and BX-C Hox proteins (Bejsovec, 2018; Bender, 2020; Gehring et al, 2009; Maeda and Karch, 2006; Nusse and Clevers, 2017; Pearson et al, 2005), it will be fascinating to determine whether their interplay similarly regulates adipocyte heterogeneity in mammals. Dissecting these mechanisms in the genetically tractable *Drosophila* model may yield valuable insights into the evolution, specialization, and pathological dysregulation of adipocytes relevant to human diseases.

## Limitations of the study

In *Drosophila* larvae, adipogenesis is completed during late embryogenesis (Hoshizaki, 2005; Rizki and Rizki, 1978), and our analyses focused on L3-wandering larvae (~ 88 h post-hatching). This timeframe limits the duration available for Gal4-UAS activation, RNAi-mediated mRNA depletion, and the observation of measurable effects on lipid metabolism. The relatively milder effects observed from the depletion of *Axn*, *abd-A*, and *Abd-B* could be attributed to the stability of mRNA and proteins or the efficiency of RNAi, potentially underestimating the full impact of their loss. We used EGFP-tagged BX-C lines to detect endogenous BX-C protein expression. However, discrepancies with antibody-based studies highlight challenges in detecting low-expression transcription factors (Banreti et al, 2014; Duffraisse et al, 2020; Marchetti et al, 2003). Variations in developmental stages, fixation methods, or imaging conditions could account for these differences. Additionally, depleting Abd-A and Abd-B in larval adipocytes delayed the larval-to-pupa transition and reduced pupal viability. The underlying cause of lethality remains unclear, as a 20% reduction in triglycerides by itself may be insufficient to explain this phenotype. Further studies are needed to elucidate the mechanisms behind these effects.

## Methods

### Reagents and tools table

| Reagent/resource | Reference or source | Identifier or catalog number |
|---|---|---|
| **Experimental models** | | |
| List of *Drosophila melanogaster* stocks | This study | Appendix Table S1 |

| Reagent/resource | Reference or source | Identifier or catalog number |
|---|---|---|
| **Recombinant DNA** | | |
| pNP vector | Qiao et al, 2018 | N/A |
| **Oligonucleotides and other sequence-based reagents** | | |
| List of Primers | This study | Appendix Table S2 |
| **Chemicals, enzymes and other reagents** | | |
| DAPI | Sigma-Aldrich | D9542 |
| BODIPY | ThermoFisher | D3922 |
| Phalloidin | ThermoFisher | B3475 |
| HCR Probe abdA-B1-488 | Molecular Instruments | PRO337 |
| HCR Probe FASN1-B1-488 | Molecular Instruments | PRQ415 |
| HCR Probe fz3-B1-488 | Molecular Instruments | RTD102 |
| HCR Probe AbdB-B2-594 | Molecular Instruments | RTB307 |
| HCR Probe Lsd2-B2-594 | Molecular Instruments | PRQ419 |
| Phusion High-Fidelity DNA Polymerase | ThermoFisher | F530L |
| TRIzol | ThermoFisher | 15596026 |
| Quick-RNA MiniPrep kit | Zymo Research | R1055 |
| TruSeq Stranded Total RNA Library Prep kit | Illumina | 20020596 |
| SYBR Green Master Mix | ThermoFisher | A46109 |
| **Software** | | |
| Flybase | | https://flybase.org/ |
| ImageJ | National Institutes of Health | https://fiji.sc/ |
| Adobe Photoshop | Adobe | https://www.adobe.com/ |
| SnapGene | | https://www.snapgene.com |
| R (v4.0.4) | R Core Team | https://www.rproject.org/ |
| IGV | | https://igv.org |
| Prism 8 | GraphPad | https://www.graphpad.com/scientific-software/prism/ |

## Methods and protocols

### Drosophila stocks and maintenance

Flies used in this study were maintained at 25 °C on a standard diet consisting of cornmeal, molasses, and yeast. Unless otherwise specified, all larvae analyzed were in the later third-instar wandering stage, which typically lasts 8–10 h and occurs approximately 108–118 h after egg laying at 25 °C (Xie et al, 2015). Our study focused on female larvae in the mid-wandering L3 stage, during which larvae remain motile and continue feeding. The $Axn^{127}/TM6B$ and $UAS$-$Axn^{RNAi}$ lines were previously described (Liu et al, 2024; Zhang et al, 2017), and the genotypes of additional strains are provided in Appendix Table S1. Using a pNP vector-based system (Qiao et al, 2018), we generated the $UAS$-$abd.A^{RNAi}$ $Abd.B^{RNAi}$ line (primer sequences are listed in Appendix Table S2) and validated through sequencing (Appendix Fig. S8).

### Cell biological analyses

Staining of the larval fat body with BODIPY, DAPI, and phalloidin was performed as previously described (Li et al, 2022; Zhang et al, 2017). Triglyceride quantifications followed the same protocols, using a colorimetric assay kit (Li et al, 2022; Zhang et al, 2017).

### The HCR RNA-FISH assay

The multiplexed in situ HCR RNA-FISH assay was conducted as described previously (Li et al, 2022). Probe sets were obtained from Molecular Instruments, including B1-Alexa Fluor 488 amplifiers targeting $abd$-$A$ (lot number PRO337), $FASN1$ (lot number PRQ415), and $fz3$ (RTD102); and B2-Alexa Fluor 594 amplifiers targeting $Abd$-$B$ (lot number RTB307) and $Lsd$-$2$ (PRQ419). Confocal images were captured using a Zeiss LSM900 confocal microscope, with representative images presented for each experiment. Quantification of HCR RNA-FISH images has been described previously (Liu et al, 2025).

### RNA-seq sample preparation, library preparation, sequencing, differential gene expression analysis, and gene ontology enrichment analysis

These experiments were conducted following previously described methods (Li et al, 2022; Liu et al, 2024; Liu et al, 2025).

### Quantitative RT-PCR

The total RNA isolation, reverse transcription, and qPCR analyses were performed as described previously (Zhao et al, 2012). Dissected fat bodies from third-instar wandering larvae (25–35 each) in three biological repeats for each genotype were analyzed. The primer sequences used in qRT-PCR assay are listed in Appendix Table S2.

### Tagging the endogenous abd-A and Abd-B loci with EGFP using CRISPR-Cas9

To generate the endogenously EGFP-tagged $abd$-$A$ (CG10325) line, the EGFP-PBacDsRed cassette was inserted, replacing the ATG of $abd$-$A$-$RB$ isoform. A guide RNA (5'-TGACGAATTCGGGGGGGGTGG[TGG]-3') directed the insertion. The cassette includes EGFP and a PBacDsRed selection marker. Homology arms flanking the target site (Appendix Fig. S6A) were amplified from genomic DNA using Phusion High-Fidelity DNA Polymerase (Thermo Scientific) under optimized conditions reflecting the injection strain's genome. The donor plasmid was constructed via Golden Gate cloning into pUC57-Kan, followed by bacterial transformation, colony PCR, and sequencing. The PBacDsRed marker, flanked by PBac terminal repeats, contains a 3xP3 promoter (three tandem copies of the Pax-6 homodimer binding site and TATA-homology of hsp70) driving DsRed2 expression, facilitating genetic screening. This marker can be excised via PiggyBac transposase, leaving a single TTAA motif. A 'Val-Lys' linker (encoded by GTTAAA) remains between EGFP and $abd$-$A$. Marker excision was confirmed using 'excision PCR' with primers targeting EGFP ('Excision-abd.A5.1') and downstream of the homology arm ('Excision-abd.A3.1'). To prevent re-cutting by Cas9, PAM site mutations (5'-TGACGAATTCGG GGGGGTGG[TCG]-3') were introduced into the donor sequence.

The same strategy was applied to tag *Abd-B-RB* (*CG11648*) (Appendix Fig. S6B), using a guide RNA (5'-AGATGGTGCTGCTGCATGAC[GGG]-3'). In this case, PAM site mutation was unnecessary. CRISPR-mediated mutagenesis was conducted by WellGenetics. Primer sequences are provided in Appendix Table S2.

### Statistical analyses

*P* values were calculated using one-tailed unpaired *t* tests, where our experimental models are based on well-defined directional predictions regarding the outcomes. Error bars in all figures represent standard deviations. Each experiment was performed with a minimum of three independent biological replicates (unless otherwise mentioned). Statistical significance is indicated as follows: $P < 0.05$ (*/#), $P < 0.01$ (**/##), and $P < 0.001$ (***/###).

## Data availability

The RNA-seq data in this study have been deposited in NCBI's Gene Expression Omnibus and are accessible through GEO Series accession number GSE280511.

The source data of this paper are collected in the following database record: biostudies:S-SCDT-10_1038-S44319-025-00625-z.

## Peer review information

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

## Acknowledgements

We thank Yashi Ahmed, Wei Du, Ingrid Lohmann, and Jian-Quan Ni for generously sharing fly strains and reagents; Keith Maggert, Michael Levine, and Tianyi Zhang for their insightful comments on the manuscript; and Tony Ip for helpful discussions. We also thank the Bloomington *Drosophila* Stock Center (NIH Grant P40OD018537) and WellGenetics for fly stocks, and the Developmental Studies Hybridoma Bank at the University of Iowa for monoclonal antibodies. This research was supported by the National Institutes of Health (GM129266 to J-YJ).

## Author contributions

**Rajitha-Udakara-Sampath Hemba-Waduge**: Conceptualization; Data curation; Formal analysis; Investigation; Writing—original draft; Writing—review and editing. **Mengmeng Liu**: Data curation; Formal analysis; Validation; Investigation; Methodology; Writing—original draft. **Xiao Li**: Resources; Data curation; Formal analysis; Investigation; Methodology; Writing—review and editing. **Jasmine L Sun**: Investigation. **Elisabeth A Budslick**: Investigation. **Sarah E Bondos**: Formal analysis; Writing—review and editing. **Jun-Yuan Ji**: Conceptualization; Formal analysis; Supervision; Funding acquisition; Writing—original draft; Project administration; Writing—review and editing.

Source data underlying figure panels in this paper may have individual authorship assigned. Where available, figure panel/source data authorship is listed in the following database record: biostudies:S-SCDT-10_1038-S44319-025-00625-z.

## Disclosure and competing interests statement

The authors declare no competing interests.

