## [Peer Review File · EMBO Reports]

Adipocyte heterogeneity regulated by the Bithorax Complex-Wnt signaling crosstalk in *Drosophila*

Rajitha-Udakara-Sampath Hemba-Waduge, Mengmeng Liu, Xiao Li, Jasmine Sun, Elisabeth Budstick, Sarah Bondos, and Jun-Yuan Ji

Corresponding author(s): Jun-Yuan Ji (ji@tulane.edu)

Review Timeline:

Submission Date:	30th Mar 25
Editorial Decision:	9th May 25
Revision Received:	21st Jul 25
Editorial Decision:	29th Aug 25
Revision Received:	8th Sep 25
Accepted:	20th Oct 25

Editor: Deniz Senyilmaz Tiebe / Kurt Weir

Transaction Report:

Dear Dr. Ji,

Thank you for transferring your manuscript to EMBO Reports, which was now seen by two referees, whose reports are copied below.

Referees express interest in the proposed interplay between BX-C and Wnt signaling. However, they also raise some concerns that need to be addressed to consider publication here. In particular,

- Given the complex role of Axin in Wnt regulation, other means need to be employed to activate Wnt signaling (referee #1, point 1). Wnt activity needs to be assessed with alternative approaches to strengthen the conclusions (referee #2, point 4).
- Functional relevance of the of the proposed Wnt-abd-A/abd-B link in development needs to be investigated (referee #1, point 2).
- The effect of developmental timing on the observed phenotypes needs to be investigated/ruled out (referee #1, point 3).
- Gal80^{ts} systems need to be employed for temporal control and MARCM based clonal analysis used be performed for testing cell-autonomous effects (referee #2, point 2).
- The proposed TCF/Pan binding to the abd-A and abd-B loci in the adipose tissue needs to be supported with CUT&RUN or ChIP-qPCR in the adipose tissue (referee #2, point 1)
- A preliminary chromatin profiling will strengthen the proposed model by which abd-A and abd-B regulate Wnt signaling (referee #2, point 8).
- Concerns related to sample purity, quantifications, statistics and the model figure need to be addressed (referee #2, points 3, 5, 6).
- Discussion section needs to be expanded to acknowledge alternative interpretations of the results (referee #2, point 7).

Please contact me if you have questions or comments regarding any of these points or the revision for further discussion (also by video chat).

Given these recommendations, we would like to invite you to revise your manuscript with the understanding that the referee concerns (as in their reports) must be fully addressed and their suggestions taken on board. Please address all referee concerns in a complete point-by-point response. Acceptance of the manuscript will depend on a positive outcome of a second round of review. It is EMBO reports policy to allow a single round of major experimental revision only and acceptance or rejection of the manuscript will therefore depend on the completeness of your responses included in the next, final version of the manuscript.

We realize that it is difficult to revise to a specific deadline. In the interest of protecting the conceptual advance provided by the work, we recommend a revision within 3 months. Please discuss the revision progress ahead of this time with me if you require more time to complete the revisions, or if you have questions or comments regarding the revision (also by video chat).

1. A data availability section providing access to data deposited in public databases is missing (where applicable).
2. Your manuscript contains statistics and error bars based on $n=2$. Please use scatter plots in these cases.

You can submit the revision either as a Scientific Report or as a Research Article. For Scientific Reports, the revised manuscript can contain up to 5 main figures and 5 Expanded View figures, and it should not exceed 27000 characters. If the revision leads to a manuscript with more than 5 main figures it will be published as a Research Article. In this case the Results and Discussion section should be separate. If a Scientific Report is submitted, these sections have to be combined. This will help to shorten the manuscript text by eliminating some redundancy that is inevitable when discussing the same experiments twice. In either case, all materials and methods should be included in the main manuscript file.

3) We replaced Supplementary Information with Expanded View (EV) Figures and Tables that are collapsible/expandable online. A maximum of 5 EV Figures can be typeset. EV Figures should be cited as 'Figure EV1, Figure EV2' etc... in the text and their respective legends should be included in the main text after the legends of regular figures.

4) a .docx formatted letter INCLUDING the reviewers' reports and your detailed point-by-point responses to their comments. As part of the EMBO publication's Transparent Editorial Process, EMBO reports publishes online a Review Process File (RPF) to accompany accepted manuscripts. This File will be published in conjunction with your paper and will include the referee reports, your point-by-point response and all pertinent correspondence relating to the manuscript.

<https://www.embopress.org/page/journal/14693178/authorguide#transparentprocess>

5) a complete author checklist, which you can download from our author guidelines

<https://www.embopress.org/page/journal/14693178/authorguide>. Please insert information in the checklist that is also reflected in the manuscript. The completed author checklist will also be part of the RPF.

6) Please note that all corresponding authors are required to supply an ORCID ID for their name upon submission of a revised manuscript (<<https://orcid.org/>>). Please find instructions on how to link your ORCID ID to your account in our manuscript tracking system in our Author guidelines

<<https://www.embopress.org/page/journal/14693178/authorguide#authorshipguidelines>>

7) Before submitting your revision, primary datasets produced in this study need to be deposited in an appropriate public database (see <https://www.embopress.org/page/journal/14693178/authorguide#datadeposition>). Please remember to provide a reviewer password if the datasets are not yet public. The accession numbers and database should be listed in a formal "Data Availability" section placed after Materials & Method (see also

<https://www.embopress.org/page/journal/14693178/authorguide#datadeposition>). Please note that the Data Availability Section is restricted to new primary data that are part of this study. * Note - All links should resolve to a page where the data can be accessed. *

Additional information on source data and instruction on how to label the files are available:

<https://www.embopress.org/page/journal/14693178/authorguide#sourcedata>

9) Our journal encourages inclusion of *data citations in the reference list* to directly cite datasets that were re-used and obtained from public databases. Data citations in the article text are distinct from normal bibliographical citations and should directly link to the database records from which the data can be accessed. In the main text, data citations are formatted as follows: "Data ref: Smith et al, 2001" or "Data ref: NCBI Sequence Read Archive PRJNA342805, 2017". In the Reference list, data citations must be labeled with "[DATASET]". A data reference must provide the database name, accession number/identifiers and a resolvable link to the landing page from which the data can be accessed at the end of the reference. Further instructions are available at <http://www.embopress.org/page/journal/14693178/authorguide#referencesformat>

10) Regarding data quantification (see Figure Legends:

<https://www.embopress.org/page/journal/14693178/authorguide#figureformat>)

12) Please also note our reference format:

13) All Materials and Methods need to be described in the main text using our 'Structured Methods' format, which is required for all research articles. According to this format, the Methods section includes a Reagents and Tools Table (listing key reagents, experimental models, software and relevant equipment and including their sources and relevant identifiers) followed by a Methods and Protocols section describing the methods using a step-by-step protocol format. The aim is to facilitate adoption of the methodologies across labs. More information on how to adhere to this format as well as a downloadable template (.docx) for the Reagents and Tools Table can be found in our author guidelines:

I look forward to seeing a revised version of your manuscript when it is ready. Please let me know if you have questions or comments regarding the revision.

Kind regards,

Deniz Senyilmaz Tiebe

Deniz Senyilmaz Tiebe, PhD
Senior Scientific Editor
EMBO Reports

Referee #1:

The manuscript entitled " Adipocyte heterogeneity regulated by the Bithorax Complex-Wnt signaling crosstalk in Drosophila" by Rajitha-Udakara-Sampath Hembra-Waduge et al., describes a series of experiments aimed at understanding the phenotype of axin mutants in the Drosophila fat body. This work led to a connection between the axin fat body phenotype and the posterior homeotic genes, abd-A and Abd-B. To summarize their results, axin mutants (or at least one allele of axin) develop into larvae in which the abdominal portion of the fat body is much less prominent. This phenotype seems to be related to a loss of function in axin, as knockdown experiments produce a similar, though less severe phenotype. Upon closer inspection of the axin RNAi phenotype, the authors find that in the posterior fat body, the adipocytes form two population, one with smaller cells and one in which the cells are extremely large and abnormally shaped (much less frequent). Based on the posterior localization and transcriptomic analysis, the authors focus on the posterior bithorax complex genes, which were previously shown to be expressed in the fat body. Axin knockdown phenotypes are suppressed by knockdown of abd-A and Abd-B, but not Ubx.

Furthermore, overexpression of Abd-A and B in more anterior segments induces the Axin phenotype more anteriorly. Based on these experiments, the authors conclude that the axin phenotype works through the posterior hox genes. They furthermore examine potential targets of these genes using transcriptomics and suggest that abd-A and B are gene involved in the activation of genes involved in fatty acid/lipid metabolism and the repression of genes involved in lipid biosynthesis. Overall, I found the manuscript full of interesting experiments, and worth consideration for publication in EMBO reports. I had a hard time deciding on how to judge its importance/general interest and I had some questions and comments that I would like addressed (or explained) prior to taking a final decision. Basically, the authors may be able to address my concerns directly and show me that this work is of high general interest. So here goes:

1. One problem I had in comprehending the manuscript was with regards to the role of Wnt signaling in this phenotype. My understanding is that although Axin primarily plays a negative role in Wnt signaling by bringing beta-catenin to the destruction complex, it may also play a positive role by bringing GSK3 to the LRP5/6 co-receptor that eventually helps to stabilize beta-catenin. While I can see that the destruction function might be epistatic to its function in activation, I found it bothering me that the authors use Axin RNAi as synonymous with Wnt activation or "active Wnt signaling". Is it not possible that Axin has other functions in Wnt signaling or outside of Wnt signaling that might cause these phenotypes? Thus, I was wondering if the authors tried other methods to activate Wnt signaling and examine the cellular phenotypes. To go along with this, I was wondering why overexpression of Wnt-4 might not show similar phenotypes?
2. I was wondering if the authors had any evidence that this pathway is used during the life of the fly? Is there a Wnt trigger at some time point or under certain environmental conditions? Because the Axin phenotype is like a gain of function mutation, it is hard to know if this phenotype shows an important function or just demonstrates what happens when a signaling cascade is activated in the wrong place. While the loss the downstream regulators Abd-A and B leads to a delay in development, this is not too dramatic and may or may not be related to the wg signaling phenotype. With regards to this, are the Arrow and/or pangolin mutants used in this study also delayed? Also, why would it be important to have this effect in the posterior FB and not more anteriorly?
3. I am also trying to integrate the old paper from the Pimpinelli group (Marchetti et al.). They show, quite clearly, Ubx expression in the fat body. The group of Merabet suggests that hox gene expression in the FB might switch at the wandering larvae stage. Thus, I wonder if the differences seen can simply be due to the timing of larval staining. This is important because of the common cross-regulatory nature of hox genes. In the ectoderm, there is a posterior down regulation rule where posterior hox genes down regulate anterior genes. When one is lost, another becomes expressed. Can you rule this out? If you were to look a little earlier, might you see expression of the more anterior hox genes and might this expression modify the cells types and your interpretations? For example, perhaps the lack of effect of Axin knockdown in the thorax is due to the expression of an anterior hox gene, that gets repressed when more posterior hox genes are expressed.
4. It seems like the axin mutant is more severe than the knockdown and affects the FB in more anterior areas. Assuming that the 127 allele is a loss of function allele, this suggests that the knockdown is incomplete and that more anterior tissues can also be affected. Regarding the incompleteness, might the knockdown of abd-A/B simply dilute the driver a bit more to suppress the phenotype? Have the authors tried to cross in mutants to check for modification of the phenotypes. If clones are too difficult maybe hemizygous conditions could be used. Regarding the more anterior tissues being affected, can the authors comment on this?
5. I like the idea that there are different cell types in the posterior FB that can be visualized by the different phenotypes brought about by Axin knockdown. It is interesting that some cells activate FB metabolism genes but that the large cells do not. In the discussion, the authors suggest that the large adipocytes do not seem to express abd-A or Abd-B based on previous experiments. If abd-a and B are able to suppress the formation large adipocytes in Axin knockdowns, why are there large cells forming in the posterior FB where these genes are expressed? Are there many cells in the posterior FB that do not express either hox gene? Are their numbers similar to the number of large adipocytes formed in Axin knockdowns? This would also suggest that formation of these cells is a cell non-autonomous event that requires the posterior hox genes in neighboring cells. If so, then one might expect to find cells just outside of the anterior border of the abd-A region also being large. Do the authors have Abd-A/B staining in wild type, and mutant FBs?

Referee #2:

Review for EMBOR-2025-61647-T

The manuscript by Hembra-Waduge et al. investigates the molecular basis of adipocyte heterogeneity in *Drosophila* larvae, focusing on the interplay between the Bithorax Complex (BX-C) Hox genes (abd-A and Abd-B) and Wnt signaling. The authors propose that abdominal adipocytes express abd-A and Abd-B, enabling them to respond differently to Wnt pathway activation compared to thoracic adipocytes, and that Wnt signaling further amplifies the expression of these Hox genes, establishing a feedforward regulatory loop. The study addresses an important topic, the regional specialization of fat tissue, and provides intriguing observations. However, the mechanistic underpinning remains largely correlative, essential controls are missing, and the key claims are overstated relative to the presented data. Overall, the manuscript offers incremental rather than

transformative insights and would require major experimental and conceptual revisions to meet the standards of EMBO Reports.

Major Points:

1. The central claim that a feedforward loop exists between Wnt signaling and BX-C genes in the larval fat body remains insufficiently supported. The CUT&RUN data cited (Fig. 6A and Supplementary Fig. S11) were generated from wing imaginal discs, not from adipocytes, yet conclusions are drawn as if they apply directly to the fat body. Given the well-established tissue specificity of chromatin landscapes, these data cannot be extrapolated to the fat body without experimental validation. The authors should perform CUT&RUN or ChIP-qPCR in adipocyte tissue to directly test for TCF/Pan binding at *abd-A* and *Abd-B* loci.
2. The experimental system relies on *Axn* mutants and fat body-specific RNAi to activate Wnt signaling, but systemic developmental effects are likely. The reliance on global mutants and late Gal4 activation (as in Figs. 1B, 1D, 2C, 2E) raises concerns about secondary effects on lipid storage. Temporal control using Gal80^{ts} should be implemented to restrict Wnt activation to specific developmental windows, and MARCM-based clonal analysis should be used to test cell-autonomous effects. Without this, the possibility that changes are indirect consequences of systemic metabolic dysfunction remains open.
3. The phenotypic analyses throughout the manuscript (especially Figs. 1, 2, 4, and 5) are qualitative. Although the figures nicely depict changes in larval transparency and adipocyte morphology, there is no systematic quantification of lipid droplet size, adipocyte size distribution, or triglyceride content. For instance, in Figures 1F' and 1H', changes in adipocyte size are visually obvious but not measured; similarly, Figures 2I-2O show rescue phenotypes but provide no quantitative validation. Quantitative analyses should be added, with appropriate statistical treatment and sufficient biological replicates.
4. The use of a single Wnt activity reporter, *fz3-RFP* (Figs. 4C-4H), is insufficient to assess Wnt pathway activation. Wnt signaling regulates different targets in different contexts, and using only *fz3-RFP* risks misinterpretation. Additional reporters (e.g., *nkd-lacZ*, *Notum-lacZ*) or direct qPCR analysis of Wnt target genes such as *Myc* and *CycD* should be included to validate the generality of the findings.
5. The RNA-seq experiments comparing thoracic and abdominal fat bodies (Fig. 1I-J, Fig. 7C-F) may be compromised by tissue contamination. While the authors mention the exclusion of some germline-expressed genes, no direct validation of sample purity was performed. Without confirmation that samples are free of gonad or gut contamination, the RNA-seq data remain difficult to interpret. Tissue purity should be assessed using specific markers, or, preferably, adipocytes should be FACS-sorted prior to sequencing.
6. The model figure (Fig. 7A) presents a feedforward loop between Wnt signaling and BX-C genes as definitive, without marking hypothetical steps. Given the correlative nature of the data, the model should be revised to depict hypothesized interactions using dashed lines or similar conventions, and the figure legend should clarify where direct evidence is lacking.
7. An alternative explanation for the observed adipocyte regionalization is that differences in lipid mobilization and gene expression arise from systemic physiological gradients or differential adipocyte survival, rather than being directly due to *abd-A/Abd-B* expression. This possibility is not considered in the manuscript. The authors should explicitly discuss alternative interpretations, particularly in the context of their data in Figures 1, 2, and 7.
8. The broader mechanistic question of how *abd-A* and *Abd-B* modulate both Wnt-activated and Wnt-repressed target genes remains unresolved. The suggestion of a "permissive role" is intriguing but speculative. No co-factors or chromatin states are identified. Preliminary chromatin profiling (e.g., ATAC-seq or CUT&RUN for histone marks) in BX-C-depleted versus control adipocytes could strengthen this model and help interpret the RNA-seq data shown in Figs. 3 and 5 more mechanistically.

Minor Points:

1. The Introduction section somewhat overstates the novelty of the study, minimizing prior work that has shown BX-C expression and function in the larval fat body (e.g., Duffraisse et al., 2020). Although the authors cite prior studies documenting BX-C gene expression and functional roles in the larval fat body, they do not adequately acknowledge the extent of this earlier work. As a result, the manuscript presents BX-C involvement in fat body biology as a novel discovery, thereby overstating the originality of their findings.
2. The discussion speculates extensively on the evolutionary origins of adipocyte heterogeneity, comparing *Drosophila* fat body cells to mammalian white and brown adipocytes (Discussion, final pages). These are interesting ideas but should be explicitly framed as hypotheses rather than conclusions.
3. RNAi knockdown efficiencies for *abd-A*, *Abd-B*, and *Axn* are not validated experimentally. Given that many of the central claims rest on RNAi-mediated depletion (e.g., Figs. 2, 4, and 5), the authors should provide qPCR or Western blot quantification of knockdown efficiency.
4. The number of biological replicates used in imaging (e.g., for Figures 2, 4, and 5) and for quantifications (e.g., Figure 5H) should be clearly indicated in each figure legend.
5. The manuscript describes statistical analysis methods only briefly. One-tailed t-tests are used throughout without proper justification. Unless the authors can rigorously justify the use of one-tailed tests based on experimental design, standard two-tailed t-tests should be employed.
6. Supplementary figures, such as the validation of EGFP-tagged lines (Supplementary Figs. S4-S5), are important for interpreting the main data but are not sufficiently discussed in the text. Clear integration of validation data into the Results section would improve clarity.

Conclusion: Major Revisions

While the manuscript presents interesting observations regarding regional specialization of adipocytes and an interaction between Wnt signaling and Hox gene expression, the study lacks direct mechanistic proof, and the interpretation of the data exceeds the strength of the evidence. Major experimental revisions - including direct testing of Wnt regulation of BX-C genes in

adipocytes, rigorous phenotypic quantification, broader Wnt reporter validation, sample purity assessment, and a more cautious presentation of the model - are required for the manuscript to meet the standards expected at EMBO Reports. I therefore recommend major revisions.

We sincerely thank the reviewers for their insightful comments and constructive suggestions, which have greatly improved the quality of our work. We provide our point-by-point responses below. Reviewer comments are shown in black, and our responses are presented in dark blue.

Referee #1: The manuscript entitled " Adipocyte heterogeneity regulated by the Bithorax Complex-Wnt signaling crosstalk in Drosophila" by Rajitha-Udakara-Sampath Hembra-Waduge et al., describes a series of experiments aimed at understanding the phenotype of axin mutants in the Drosophila fat body. This work led to a connection between the axin fat body phenotype and the posterior homeotic genes, *abd-A* and *Abd-B*. To summarize their results, axin mutants (or at least one allele of axin) develop into larvae in which the abdominal portion of the fat body is much less prominent. This phenotype seems to be related to a loss of function in axin, as knockdown experiments produce a similar, though less severe phenotype. Upon closer inspection of the axin RNAi phenotype, the authors find that in the posterior fat body, the adipocytes form two population, one with smaller cells and one in which the cells are extremely large and abnormally shaped (much less frequent). Based on the posterior localization and transcriptomic analysis, the authors focus on the posterior bithorax complex genes, which were previously shown to be expressed in the fat body. Axin knockdown phenotypes are suppressed by knockdown of *abd-A* and *Abd-B*, but not *Ubx*. Furthermore, overexpression of *Abd-A* and *B* in more anterior segments induces the Axin phenotype more anteriorly. Based on these experiments, the authors conclude that the axin phenotype works through the posterior hox genes. They furthermore examine potential targets of these genes using transcriptomics and suggest that *abd-A* and *B* are gene involved in the activation of genes involved in fatty acid/lipid metabolism and the repression of genes involved in lipid biosynthesis. Overall, I found the manuscript full of interesting experiments, and worth consideration for publication in EMBO reports. I had a hard time deciding on how to judge its importance/general interest and I had some questions and comments that I would like addressed (or explained) prior to taking a final decision. Basically, the authors may be able to address my concerns directly and show me that this work is of high general interest. So here goes:

1. One problem I had in comprehending the manuscript was with regards to the role of Wnt signaling in this phenotype. My understanding is that although Axin primarily plays a negative role in Wnt signaling by bringing beta-catenin to the destruction complex, it may also play a positive role by bringing GSK3 to the LRP5/6 co-receptor that eventually helps to stabilize beta-catenin. While I can see that the destruction function might be epistatic to its function in activation, I found it bothering me that the authors use Axin RNAi as synonymous with Wnt activation or "active Wnt signaling". Is it not possible that Axin has other functions in Wnt signaling or outside of Wnt signaling that might cause these phenotypes? Thus, I was wondering if the authors tried other methods to activate Wnt signaling and examine the cellular phenotypes. To go along with this, I was wondering why overexpression of Wnt-4 might not show similar phenotypes?

Response: We appreciate your thoughtful questions and insightful comments. Regarding the effect of *Axn* depletion on Wnt signaling, our previous publications have demonstrated that RNAi-mediated knockdown or mutation of *Axn* (e.g., the *Axn*¹²⁷ allele) leads to stabilization of Armadillo/Arm, the *Drosophila* ortholog of β -catenin. This leads to elevated Wnt target gene expression and reduced lipid accumulation (Zhang et al., 2017/PMID: 28827348/<https://pubmed.ncbi.nlm.nih.gov/28827348>; Liu et al., 2024/PMID: 38968125/<https://www.pnas.org/doi/10.1073/pnas.2322066121>).

Specifically, we observed that: (a) the *Axn*¹²⁷ allele disrupts the C-terminal DIX domain of Axn, essential for its oligomerization with Dishevelled. This mutation causes elevated Arm levels and increased expression of Wnt-activated genes such as *naked cuticle* (*nkd*) and *Notum* (Zhang et al., 2017/PMID: 28827348). (b) RNAi-knockdown of either *Axn* or *slmb* (*supernumerary limbs*, encodes the *Drosophila* ortholog of β -TrCP, an E3 ligase involved in Arm degradation) leads to increased Arm levels and activation of Wnt target genes in both larval adipocytes and the intestine (Liu et al., 2024/PMID: 38968125). (c) We have also employed additional strategies to activate Wnt signaling, including RNAi depletion of *slmb* (Liu et al., 2024/PMID: 38968125) and ectopic expression of a dominant negative form of *Xenopus* GSK3 β (*UAS-GSK3 β ^{DN}*) (Zhang et al., 2017/PMID: 28827348). These genetic manipulations produced similar effects on Arm stabilization, Wnt target gene expression, and lipid homeostasis. (d) Furthermore, treatment of cultured S2R+ cells with Wingless/Wg-conditioned medium also reduced fat and lipid droplet accumulation, similar to the effects observed upon *Axn* or *slmb* depletion in larval adipocytes (Liu et al., 2024/PMID: 38968125). Consistently, overexpression of *wg* in the larval fat body also led to reduced fat accumulation (Lee et al., 2014/PMID: 24979807). (e) Conversely, depletion of Arm or Pan/dTCF increased lipid accumulation, opposite to the effects seen with *Axn* or *slmb*

knockdown (Liu et al., 2024/PMID: 38968125). Taken together, these findings strongly support the role of Axn as a negative regulator of Arm and argue against the possibility that the observed phenotypes arise from Axn functions unrelated to Wnt signaling.

Regarding the lack of observable effects from Wnt4 overexpression in larval fat body: Prior studies have shown that Wnt4 antagonizes Wg/Wnt1 signaling in the ventral ectoderm of *Drosophila* embryos and suppresses the axis-inducing activity of Wg in *Xenopus* embryos (Gieseler et al., 1999/PMID: 10415353). Ectopic expression of Wnt4 was also found to repress *wg* expression and moderately reduce Arm levels in stage 11 embryos (Gieseler et al., 1999/PMID: 10415353). Therefore, overexpressing Wnt4 in larval adipocytes is unlikely to activate canonical Wnt signaling. Moreover, the *Drosophila* larval stage spans approximately four days. It is possible that Wnt overexpression during this period is insufficient to have a significant impact on lipid metabolism. In the revised manuscript, we now cite Gieseler et al., 1999 (PMID: 10415353) to clarify this point. As noted earlier, overexpression of *wg* in larval fat body using *dCg-Gal4* resulted in reduced fat accumulation, particularly in the abdominal region (Lee et al., 2014/PMID: 24979807), consistent with our previous findings and the current work. Additionally, depletion of Wnt4 did not result in obvious changes in lipid metabolism (Fig. S2C), further supporting our interpretation. Based on these findings, we did not pursue further analysis of Wnt4.

2. I was wondering if the authors had any evidence that this pathway is used during the life of the fly? Is there a Wnt trigger at some time point or under certain environmental conditions? Because the Axin phenotype is like a gain of function mutation, it is hard to know if this phenotype shows an important function or just demonstrates what happens when a signaling cascade is activated in the wrong place. While the loss of the downstream regulators Abd-A and B leads to a delay in development, this is not too dramatic and may or may not be related to the *wg* signaling phenotype. With regards to this, are the Arrow and/or pangolin mutants used in this study also delayed? Also, why would it be important to have this effect in the posterior FB and not more anteriorly?

Response: In our recent paper (Liu et al., 2024/PMID: 38968125), we reported that reducing Wnt signaling – via depletion of key downstream effectors such as *arm/β-Catenin* or *pan/dTCF*, or by overexpressing *Notum* (a Wnt signaling antagonist) or a dominant-negative form of dTCF (*dTCF^{DN}*) – more effectively revealed its physiological role in lipid metabolism. These findings suggest that Wnt signaling is active in the fat body under normal conditions, albeit at low levels.

Your question regarding the trigger of Wnt signaling is both important and challenging. To our knowledge, the source of Wnt ligands regulating lipid homeostasis in adipocytes, as well as the environmental or physiological cues that modulate them, remains unknown. *Drosophila* encodes seven Wnt paralogs (*Wg*, *Wnt2*, *Wnt4*, *Wnt5*, *Wnt6*, *Wnt10*, and *WntD*), which may be secreted from other tissues or may diffuse through the hemolymph to act as inter-organ signaling molecules. Further studies are needed to elucidate the origins and regulation of these Wnt ligands.

Regarding the developmental delay in larvae with *abd-A* or *Abd-B* depletion: we agree that the delay is modest and may not be directly linked to Wnt signaling. Abd-A and Abd-B regulate many target genes, some of which are likely unrelated to Wnt signaling in the fat body. Since we did not observe obvious developmental delays in larvae with fat body-specific depletion of Arrow, Pan, or Arm, this phenotype was not further pursued.

As for the significance of regional differences in the larval fat body, this is a fascinating and complex question that we have been considering for some time. Although a definitive answer remains elusive, we propose three speculative scenarios: (a) Metabolic transport: Thoracic adipocytes in wild-type larvae display prominent kynurenine autofluorescence (KAF⁺) compared to abdominal adipocytes (Butterworth et al., 1988/PMID: 3136556; Rizki, 1961/PMID:13741990). Since tryptophan and kynurenine are precursors of the eye pigment ommochrome, and thoracic adipocytes are physically closer to eye imaginal discs, they may help shuttle these metabolites to the discs during metamorphosis (Rizki and Rizki, 1972. *Egyptian Journal of Genetics and Cytology* 1, 173-188). (b) Adipocyte heterogeneity: in mammals, adipocytes from different adipose depots are linked to different metabolic diseases, though the molecular basis remains poorly understood. The regional difference we observed in this work may represent an evolutionarily primitive form of such adipocyte heterogeneity. (c) Structural and developmental roles: The ventral arm of the thoracic fat body contacts and cushions organs, including the salivary glands, central nervous system (brain and ventral nerve cord), and imaginal discs (labial, eye, and leg discs). Preferential use of abdominal lipid stores may meet urgent energy needs (e.g., during wandering or pupariation) while preserving lipid stores in regions critical for

structural or developmental functions. Notably, these functional and regional differences in insect fat body have been further discussed by Haunerland and Shirk (*Annual Review of Entomology* 40, 121-145, 1995), and similar principles may apply to the *Drosophila* larvae fat body. It is hoped that our model (Fig. 7A) will serve as a testable hypothesis to elucidate the underlying molecular mechanisms of adipocyte heterogeneity in other insects and mammals.

3. I am also trying to integrate the old paper from the Pimpinelli group (Marchetti et al.). They show, quite clearly, Ubx expression in the fat body. The group of Merabet suggests that hox gene expression in the FB might switch at the wandering larvae stage. Thus, I wonder if the differences seen can simply be due to the timing of larval staining. This is important because of the common cross-regulatory nature of hox genes. In the ectoderm, there is a posterior down regulation rule where posterior hox genes down regulate anterior genes. When one is lost, another becomes expressed. Can you rule this out? If you were to look a little earlier, might you see expression of the more anterior hox genes and might this expression modify the cells types and your interpretations? For example, perhaps the lack of effect of Axin knockdown in the thorax is due to the expression of an anterior hox gene, that gets repressed when more posterior hox genes are expressed.

Response: Marchetti et al. (2003/PMID: 12835385/<https://pubmed.ncbi.nlm.nih.gov/12835385/>) used monoclonal and polyclonal antibodies to detect BX-C proteins, whereas we used EGFP-tagged endogenous Abd-A, Abd-B (this work), and Ubx (Domech et al. 2019/PMID: 31050646). Both approaches have their advantages and limitations. Immunostaining enables signal amplification through secondary antibodies, but it may lack the specificity of endogenously tagged EGFP proteins. In contrast, EGFP tagging provides high specificity without signal amplification. For instance, although the studies (Banreti et al. 2014/PMID: 24389064/<https://doi.org/10.1016/j.devcel.2013.11.024>) and Marchetti et al. (2003) used the same monoclonal FP3.38-DSHB antibody for Ubx, they observed different expression patterns. Banreti et al. (2014) detected Ubx signal strongly in fat body regions 5 and 6 (according to Riziki, T.M., 1978, Fat body. in *The genetics and biology of Drosophila*, ed. M Ashburner, Wright, T.R.F., pp. 561-601. Academic Press, Inc.) (posterior to the gonads) in male larvae, while Marchetti et al. (2003/PMID: 12835385) reported Ubx expression only in the abdominal region. The reasons for these differences could be due to varying fixation conditions, antibody dilution rates, antibody specificities, and confocal imaging settings.

Considering these technical factors, we chose a CRISPR-Cas9 strategy to tag the endogenous *abd-A* and *Abd-B* loci. Using EGFP-tagged lines, we observed clear nuclear localization of Abd-A and Abd-B only in abdominal adipocytes of wandering-stage larvae (Fig. 1M-P). Specifically: (a) Ubx^{EGFP} was undetectable in the L3 larval fat body at both feeding and wandering stages (Supplementary Fig. 3A-F); robust expression was observed in the larval nerve cord and haltere discs, validating the reporter. In addition, *in vitro* EGFP-Ubx is both stable and soluble (Tsai et al., 2015; <https://doi.org/10.1002/adfm.201402997>). (b) Abd-A^{EGFP} and Abd-B^{EGFP} were restricted to abdominal adipocytes, with Abd-B^{EGFP} further localized to the posterior abdominal fat body. (c) While Abd-A^{EGFP} expression slightly declined from the feeding to wandering stages, Abd-B^{EGFP} levels remained stable (Supplementary Fig. 5). These findings differ from those reported by Marchetti et al., likely due to differences in detection methodology. Given the limitations associated with antibody-based staining, we currently favor the specificity and reproducibility of EGFP-tagged endogenous proteins. We are continuing to examine these patterns and are preparing a follow-up manuscript focused on the mechanisms underlying the heterogeneous *abd-A* and *Abd-B* expression in larval adipocytes, including further analysis of *Ubx* expression.

On the potential role of posterior prevalence of *Hox* genes in larval fat body: This is an interesting question. However, our RNA-seq data do not support this model. As shown in Fig. 3B, simultaneous depletion of *abd-A* and *Abd-B* does not lead to upregulation of *Ubx* or *Antp*, as would be expected if posterior prevalence were active in the larval fat body. These findings argue against the operation of posterior prevalence in the regulation of Hox gene expression in the larval fat body.

Regarding your suggestion that an anterior *Hox* gene in thorax may be repressed by posterior Hox genes in the abdominal fat body: As shown in Fig. 2B-G, ectopic expression of *Abd-A* or *abd-B* in an *Axn*-depleted background induced adipocyte heterogeneity in the thoracic fat body, phenocopying the abdominal fat body. These results suggest that the absence of *abd-A* and *abd-B* expression in thoracic adipocytes is a key determinant of their distinct response to Wnt signaling. Further supporting this, depletion of *abd-A*, *Abd-B*, or both in abdominal adipocytes suppressed Wnt-induced adipocyte heterogeneity in the abdominal fat body (Fig. 2K/M/O). These findings established the causal relationship between Abd-A/B and Wnt signaling in regulating regional differences in lipid homeostasis in the fat body.

4. It seems like the axin mutant is more severe than the knockdown and affects the FB in more anterior areas. Assuming that the 127 allele is a loss of function allele, this suggests that the knockdown is incomplete and that more anterior tissues can also be affected. Regarding the incompleteness, might the knockdown of *abd-A/B* simply dilute the driver a bit more to suppress the phenotype? Have the authors tried to cross in mutants to check for modification of the phenotypes. If clones are too difficult maybe hemizygous conditions could be used. Regarding the more anterior tissues being affected, can the authors comment on this?

Response: As we previously reported (Zhang et al., 2017/PMID: 28827348), *Axn*¹²⁷ is a hypomorphic allele that disrupts the C-terminal DIX domain of the Axn protein but does not represent a complete loss-of-function. Complete loss of *Axn* results in embryonic lethality. Nonetheless, you are correct that the fat body phenotype observed in *Axn*¹²⁷ homozygous larvae is more severe than that seen with fat body-specific *Axn* RNAi knockdown (Zhang et al., 2017/PMID: 28827348; compare Fig. 1B vs. 1D in the current manuscript). Technically, testing genetic rescue of *Axn*¹²⁷ using classical *abd-A* and *Abd-B* alleles is considerably more time-consuming than using the GAL4-UAS system employed in our study (Fig. 2), primarily because *Axn*, *abd-A*, and *Abd-B* are all located on the third chromosome. Furthermore, all these genes are essential for embryogenesis – hemizygous mutants are embryonically lethal – while our analyses focus on their functions in the larval fat body.

Regarding the concern that “the knockdown of *abd-A/B* simply dilute the driver a bit more to suppress the phenotype”: we consider this explanation unlikely. The abundance of Gal4 proteins (*dCg-Gal4*) is generally sufficient to activate UAS-linked shRNAs simultaneously. If Gal4 dilution were responsible for the observed rescue, we would expect similar effects from both (*Axn*^{RNAi} + *abd-A*^{RNAi}) and (*Axn*^{RNAi} + *Abd-B*^{RNAi}) combinations. However, this is not what we observed. Rescue effects were predominantly detected with (*Axn*^{RNAi} + *Abd-A*^{RNAi}), not with (*Axn*^{RNAi} + *Abd-B*^{RNAi}), suggesting that the rescue is specific and gene-dependent rather than an artifact of Gal4 titration (see Supplementary Fig. 8). Conversely, overexpression of *abd-A* or *Abd-B* enhanced the *Axn*^{RNAi} phenotypes in thoracic fat body (Fig. 2E/G), further arguing against the possibility that Gal4 availability is a limiting factor in this approach.

As for the observation that more anterior adipocytes are affected in *Axn*¹²⁷ than in *Axn*^{RNAi} larvae: this can be explained by our proposed model (Fig. 7A). The boundary between thoracic and abdominal adipocytes is not sharply defined. Instead, the expression of *abd-A* and *Abd-B* in this transitional region is dynamic and regulated by intrinsic mechanisms that repress their expression in thoracic adipocytes while promoting it in abdominal adipocytes. In *Axn*¹²⁷, where Wnt signaling is more strongly activated than in *Axn*^{RNAi} larvae, this elevated Wnt signaling further enhances *abd-A* and *Abd-B* transcription, expanding their expression anteriorly. This expanded *Abd-A/B* expression enables Wnt signaling to induce lipid mobilization in more anterior regions, leading to the broader transparent fat body phenotype observed in *Axn*¹²⁷ larvae. We have accumulated additional evidence showing that *abd-A* and *Abd-B* expression undergoes tight transcriptional regulation in larval adipose tissue, and we are preparing a follow-up manuscript to describe these findings in greater detail.

5. I like the idea that there are different cell types in the posterior FB that can be visualized by the different phenotypes brought about by Axin knockdown. It is interesting that some cells activate FB metabolism genes but that the large cells do not. In the discussion, the authors suggest that the large adipocytes do not seem to express *abd-A* or *Abd-B* based on previous experiments. If *abd-a* and *B* are able to suppress the formation large adipocytes in Axin knockdowns, why are there large cells forming in the posterior FB where these genes are expressed? Are there many cells in the posterior FB that do not express either *hox* gene? Are their numbers similar to the number of large adipocytes formed in Axin knockdowns? This would also suggest that formation of these cells is a cell non-autonomous event that requires the posterior *hox* genes in neighboring cells. If so, then one might expect to find cells just outside of the anterior border of the *abd-A* region also being large. Do the authors have *Abd-A/B* staining in wild type, and mutant FBs?

Response: Not all abdominal adipocytes express *Abd-A* and *Abd-B*. Our conclusion that the large abdominal adipocytes lack *Abd-A* and *Abd-B* expression is based on the observations presented in Fig. 6I/I', Fig. 6K/K', Fig. 1M-P, and Suppl. Fig. S13F/F' and S13H/H', which collectively show their expression at both the protein and mRNA levels. These occasional large abdominal adipocytes resemble thoracic adipocytes in their absence of *Abd-A* and *Abd-B*, both of which are required for Wnt signaling to promote lipid mobilization (this study; Liu et al., 2024/PMID: 38968125). The “cells just outside of the anterior border of the *abd-A* region” correspond to thoracic adipocytes, which do not respond to Wnt/Wg signaling and remain large (i.e., non-transparent).

Regarding the number of abdominal adipocytes lacking *abd-A* and *Abd-B* expression: we have not conducted a comprehensive quantification due to technical challenges associated with the HCR method, which involves multiple rinse steps that can compromise the structural integrity of the larval fat body. Nonetheless, we estimate that these large adipocytes constitute less than ~3% of total abdominal adipocytes. This frequency aligns with the incidence of large adipocytes observed in *Axin* knockdowns and with Wnt signaling activity, as reflected by fz3-RFP (Fig. 4D/D'), *FASN*, and *Lsd-2* expression (Fig. 5E/E').

Regarding the possibility of cell-autonomous versus non-cell-autonomous expression of *abd-A/Abd-B*: the presence of occasional adipocytes that lack expression of *abd-A/Abd-B* supports the notion that their expression is in a cell-autonomous manner. In addition, we have obtained evidence indicating that *abd-A* and *Abd-B* transcription is actively regulated by several transcription factors in the larval fat body. These findings will be presented in a forthcoming manuscript. The expression of *Abd-A*^{EGFP} and *Abd-B*^{EGFP} is shown in Fig. 1M-1P.

Referee #2:

Review for EMBOR-2025-61647-T: The manuscript by Hembra-Waduge et al. investigates the molecular basis of adipocyte heterogeneity in *Drosophila* larvae, focusing on the interplay between the Bithorax Complex (BX-C) Hox genes (*abd-A* and *Abd-B*) and Wnt signaling. The authors propose that abdominal adipocytes express *abd-A* and *Abd-B*, enabling them to respond differently to Wnt pathway activation compared to thoracic adipocytes, and that Wnt signaling further amplifies the expression of these Hox genes, establishing a feedforward regulatory loop. The study addresses an important topic, the regional specialization of fat tissue, and provides intriguing observations. However, the mechanistic underpinning remains largely correlative, essential controls are missing, and the key claims are overstated relative to the presented data. Overall, the manuscript offers incremental rather than transformative insights and would require major experimental and conceptual revisions to meet the standards of EMBO Reports.

Response: We appreciate your insightful comments and critiques. This work is a follow-up study to our recently published work (Liu et al., 2024/PMID: 38968125), which provides a more mechanistic model regarding the molecular mechanisms by which active Wnt signaling stimulates lipolysis but inhibits *de novo* lipogenesis in larval adipocytes. In that study, the focus was largely on abdominal adipocytes. In contrast, this work under review focuses on the curious regional difference, i.e., the abdominal vs. thoracic adipocytes, in response to active Wnt signaling. We thank you for your appreciation of the importance of this topic. Our model is conceptually novel in the following aspects: (1) this work offers a genetic circuit to explain the regional difference in *Drosophila* larval fat body, such a regional differences have long been documented in fat body of other insects as well but with limited understanding of the molecular mechanisms that define this regional difference (Hauerland and Shirk (1995) *Annual Review of Entomology* 40, 121-145). (2) The genetic circuit regarding the interplay between Wnt signaling and Hox genes *abd-A* and *Abd-B* is novel, particularly the requirement of *Abd-A/B* on Wnt target gene expression in larval adipocytes (Fig. 7A). (3) We agree that our genetic manipulation of Wnt signaling and Hox genes allowed us to establish correlations among these factors. However, our genetic analyses (e.g., data shown in Fig. 2) and subsequent analyses enabled us to establish a causal relationship among these factors, i.e., these correlations are not simple correlations. These causal relationships are further strengthened by our RNA-seq analyses and analysis on lipid metabolism and gene expression using HCR and reporter assays. We have revised the manuscript by incorporating additional data and carefully ensuring that our conclusions are clearly conveyed. We hope our response will address your concerns, detailed below, and enable this work to meet the high standards of *EMBO Reports*.

Major Points:

1. The central claim that a feedforward loop exists between Wnt signaling and BX-C genes in the larval fat body remains insufficiently supported. The CUT&RUN data cited (Fig. 6A and Supplementary Fig. S11) were generated from wing imaginal discs, not from adipocytes, yet conclusions are drawn as if they apply directly to the fat body. Given the well-established tissue specificity of chromatin landscapes, these data cannot be extrapolated to the fat body without experimental validation. The authors should perform CUT&RUN or ChIP-qPCR in adipocyte tissue to directly test for TCF/Pan binding at *abd-A* and *Abd-B* loci.

Response: We agree that demonstrating the direct binding of dTCF/Pan on the *abd-A* and *Abd-B* loci would further strengthen our model. In response to this concern, we have included CUT&RUN data from larval adipocyte nuclei in the revised manuscript (Suppl. Fig. S12). Regarding the comparison between CUT&RUN data from wing discs and larval fat body: We originally presented the wing disc data (Liu, M. et al., 2024/PMID: 38968125) due to technical challenges encountered when working with larval fat body tissue. First, ChIP-qPCR using isolated fat body nuclei yielded an insufficient amount of DNA. Second, although we attempted CUT&RUN using dissected larval fat bodies and purified nuclei from these tissues, the resulting data were of low quality with a high noise-to-signal ratio. This may be attributed to the abundance of lipid droplets in larval adipocytes and suboptimal experimental conditions that require further refinement.

Despite these limitations, we now include CUT&RUN data from both wing discs and purified larval adipocyte nuclei to demonstrate dTCF/Pan binding at the *abd-A* and *Abd-B* loci. As shown in the revised Suppl. Fig. S12, several dTCF/Pan-binding peaks are detectable in adipocytes, overlapping with those observed in wing discs (indicated by arrows), although the adipocyte data have a higher background. Together with data presented in Fig. 6, these results provide further support for the direct regulation of *abd-A* and *Abd-B* transcription by Wnt signaling.

2. The experimental system relies on *Axn* mutants and fat body-specific RNAi to activate Wnt signaling, but systemic developmental effects are likely. The reliance on global mutants and late Gal4 activation (as in Figs. 1B, 1D, 2C, 2E) raises concerns about secondary effects on lipid storage. Temporal control using Gal80^{ts} should be implemented to restrict Wnt activation to specific developmental windows, and MARCM-based clonal analysis should be used to test cell-autonomous effects. Without this, the possibility that changes are indirect consequences of systemic metabolic dysfunction remains open.

Response: We appreciate your insightful comments regarding the potential for unknown “secondary effects” or “systematic developmental effects” on lipid storage in our Wnt activation system. While we cannot completely rule out such systematic influences, several lines of evidence suggest that their impact, if any, is minimal. Notably, similar reductions in lipid storage were observed in both *Axn*¹²⁷ mutants and in fat body-specific *Axn* knockdown using at least three different Gal4 lines (*dCg-Gal4*, *SREBP-Gal4*, and *r4-Gal4* lines), as reported in our previous studies (Zhang et al., 2017/PMID: 28827348/<https://pubmed.ncbi.nlm.nih.gov/28827348/>; Liu et al., 2024/PMID: 38968125/<https://www.pnas.org/doi/10.1073/pnas.2322066121>). Furthermore, consistent phenotypes were also observed upon *slmb* depletion (Liu et al., 2024/PMID: 38968125) or overexpression of *wingless/wg* in the larval fat body using *dCg-Gal4* (Fig. 5D in Lee et al., 2014/PMID: 24979807), both of which led to reduced fat accumulation, particularly in the abdominal region – closely resembling the phenotypes seen in *Axn*¹²⁷ mutant or fat body-specific *Axn* knockdown larvae. These findings collectively support a specific and robust role for Wnt/Wg signaling pathway activation in regulating fat storage in the larval fat body.

Regarding the cell-autonomous effects of Wnt signaling on lipid storage: We previously addressed this question in our study by generating trans-flp clones of *Axn* mutant alleles (*Axn*¹²⁷ and *Axn*^{S044230}) in the larval fat body (Suppl. Fig. S6 in Zhang et al., 2017/PMID: 28827348). We observed a marked reduction in fat accumulations specifically within *Axn* mutant clones, but not in the surrounding wild-type adipocytes (Zhang et al., 2017/PMID: 28827348). This finding supports a cell-autonomous role for Wnt signaling in regulating lipid metabolism. To avoid confounding effects of temperature, we chose not to use Gal80^{ts} system, which requires temperature shifts. Since temperature significantly influences various metabolic processes, RNAi efficiency based on the Gal4-UAS system, and developmental timing, its use would have introduced additional variables and complexity into our analyses of Wnt signaling in regulating lipid homeostasis.

3. The phenotypic analyses throughout the manuscript (especially Figs. 1, 2, 4, and 5) are qualitative. Although the figures nicely depict changes in larval transparency and adipocyte morphology, there is no systematic quantification of lipid droplet size, adipocyte size distribution, or triglyceride content. For instance, in Figures 1F' and 1H', changes in adipocyte size are visually obvious but not measured; similarly, Figures 2I-2O show rescue phenotypes but provide no quantitative validation. Quantitative analyses should be added, with appropriate statistical treatment and sufficient biological replicates.

Response: As we have explained earlier (please see Reviewer 1- major comment 4), Wnt signaling is more strongly activated in *Axn*¹²⁷ than in *Axn*^{RNAi} larvae. Therefore, *Axn*¹²⁷ results in a drastic reduction in average adipocyte sizes compared to *Axn*^{RNAi} (Fig. R1E, next page). These effects are evident when we repeat the quantifications, considering only the Wnt-active small adipocyte population (Fig. R1F). Also, we have included

Fig. R1 Adipocyte size quantifications for *Axn¹²⁷* and *Axn^{RNAi}*. Four panels (A-D) from Fig.1 E'-H' are shown together with respective adipocyte size quantifications considering all adipocytes (E) and Wnt active small adipocytes (F) in *Axn¹²⁷* and *Axn^{RNAi}*. $p < 0.05$ (*), $p < 0.01$ (**), and $p < 0.001$ (***)

whereas the adipocyte size distribution is rather evident in the confocal images, revealing that the activation of Wnt signaling can cause the adipocyte population to be heterogeneous, with the majority consisting of small adipocytes.

4. The use of a single Wnt activity reporter, *fz3-RFP* (Figs. 4C-4H), is insufficient to assess Wnt pathway activation. Wnt signaling regulates different targets in different contexts, and using only *fz3-RFP* risks

Fig. R2 Systematic quantification of adipocyte sizes upon depleting *abd-A* and *Abd-B* under Wnt-activated background. Adipocyte quantification considering all adipocytes (A) and Wnt-active small adipocytes (A') in *Axn^{RNAi}*, respectively. Different genetic combinations of *abd-A*, *Abd-B* and *Axn* depletions are shown below. $p < 0.05$ (*), $p < 0.01$ (**), and $p < 0.001$ (***)

5. The RNA-seq experiments comparing thoracic and abdominal fat bodies (Fig. 1I-J, Fig. 7C-F) may be compromised by tissue contamination. While the authors mention the exclusion of some germline-expressed genes, no direct validation of sample purity was performed. Without confirmation that samples are free of gonad or gut contamination, the RNA-seq data remain difficult to interpret. Tissue purity should be assessed using specific markers, or, preferably, adipocytes should be FACS-sorted prior to sequencing.

adipocyte size (Fig. R2) and TG level quantifications (Fig. R3) to further strengthen the genetic rescue effect, which was shown in Fig. 2I-O. We observed that Wnt-derived low TG accumulation levels are partially rescued upon concomitant depletion of *abd-A* and *Abd-B*. However, the TG levels were quantified using a kit that measures the glycerol concentration after the conversion of TGs, diacylglycerides, and monoacylglycerides into FFAs and glycerol; thus, the sensitivity of this method might be insufficient for accurately gauging the rescue effects in this experiment. In addition, we have systematically quantified the *fz3-RFP* expression levels upon Wnt activation with or without depletion of *abd-A* and *Abd-B*, as shown in revised Fig. 4G.

As the major focus of this manuscript is to study the adipocyte heterogeneity regulated by the Bithorax Complex-Wnt signaling crosstalk, we believe that the systematic quantification of lipid droplet sizes or adipocyte size distribution would not alter or significantly impact our current conclusions. In fact, accurately measuring lipid droplet sizes would require a significant effort,

Additional reporters (e.g., *nkd-lacZ*, *Notum-lacZ*) or direct qPCR analysis of Wnt target genes such as *Myc* and *CycD* should be included to validate the generality of the findings.

Response: This is an excellent question. To address this concern, we performed RT-qPCR on dissected larval fat bodies from relevant genotypes, focusing on key Wnt target genes such as *CycD*, *nkd*, and *Notum*. The results, now presented in the revised Fig. 4H, are consistent with the findings from the *fz3-RFP* reporter and provide additional support for our conclusions. While these RT-qPCR and reporter assays are informative, we believe that future RNA-seq analysis of these samples would provide a more comprehensive and unbiased view of transcriptional changes involved.

Fig. R3 Relative triglyceride levels in larvae with *abd-A/Abd-B* and/or *Axn* depletion in the larval fat body. Triglyceride (TG) levels were measured using a commercial TG quantification kit. These cells are typically 30-60 μm in diameter at the third instar larval stage, and even with a 130 μm nozzle, sorting cells larger than $\sim 50 \mu\text{m}$ remains problematic. Such attempts often result in cell rupture, clogging, low recovery, and poor viability. Therefore, standard FACS is not well-suited for isolating *Drosophila* larval adipocytes without specialized equipment and optimized protocols (e.g., gentle fluidics, 130 μm nozzle, and low pressure). Each larva has 2100-2500 adipocytes. We have also attempted to isolate nuclei from the larval fat body, but we were unable to obtain enough nuclei for downstream analyses.

Importantly, at the third instar larval stage, the larval fat body forms a single, continuous layer of interconnected adipocytes, which can be dissected in its entirety. This provides a relatively pure tissue sample, although the ovaries remain embedded in the abdominal region.

To further validate the purity of our RNA-Seq datasets, we cross-referenced our data with nine genes known to be enriched in the larval intestine, as identified using the FlyAtlas2 database (<https://flyatlas.gla.ac.uk/FlyAtlas2/index.html>).

These include *CG16723*, *Muc68E*, *CG18404*, *Bace*, *CG43187*, *LManIII*, *CG6996*, *CG43348*, and *Mip*. As shown in **Fig. R4A** and **R4B**, these genes were undetectable or barely detectable due to extremely low counts in our RNA-seq analyses using dissected fat bodies. In contrast, the expression of typical adipocyte-enriched genes such as *FASN1*, *Lsd-1*, *Lsd-2*, and *whd* was robust and consistent, arguing against intestinal contamination in our samples.

Response: The RNA-seq analysis was conducted over six years ago, at a time when it was unclear whether such an approach would yield meaningful insights. As a result, we aimed to keep the experiments straightforward, and we did not have access to a microdissection system. In hindsight, removal of the gonads may have been beneficial. However, at the onset of this project, we could not rule out the possibility that gonad-derived signals might contribute to the regional differences observed in the larval fat body. Nevertheless, based on the extensive subsequent analyses presented in Figs. 2-7 and the supplementary figures, it is highly unlikely that gonadal gene expression accounts for the regional differences among larval adipocytes. Moreover, such a model fails to explain the full range of results presented in this study. Instead, our findings strongly support a model in which crosstalk between *Abd-A/B* and *Wnt* signaling defines these regional differences.

Regarding the suggestion to use FACS for sorting larval adipocytes: this approach is technically challenging. Due to their high lipid content, large larval adipocytes tend to float, making them difficult to process on standard FACS instruments. These cells are typically 30-60 μm in diameter at the third instar larval stage, and even with a 130 μm nozzle, sorting cells larger than $\sim 50 \mu\text{m}$ remains problematic. Such attempts often result in cell rupture, clogging, low recovery, and poor viability. Therefore, standard FACS is not well-suited for isolating *Drosophila*

Fig. R4 Bar charts showing feature counts from RNA-seq data for adipocyte-expressed and larval midgut-specific genes. Expression levels are shown for four adipocyte-enriched genes (*FASN1*, *Lsd-1*, *Lsd-2*, *whd*) and nine midgut-enriched genes that were detected in RNA-seq datasets derived from dissected fat bodies. (A) Feature counts from the *dCg>Gal4/+* control sample in the *dCg>AiBi* dataset. (B) Feature counts from wild-type thoracic (WT-Th) and wild-type abdominal (WT-Ab) fat body samples. The read counts from RNA-seq analyses (three biological replicates) are shown above each of the nine genes enriched in the larval intestine.

More importantly, our genetic analyses (e.g., Fig. 2 and other related data summarized in this study) revealed *causal relationships* among the key regulatory factors under investigation. These findings further support the integrity of our RNA-seq data and argue against the possibility that the regulatory network depicted in Fig. 7A is an artifact of sample contamination or tissue impurity. It is difficult to see how random contamination could account for the coherent and consistent results presented in Figs. 2-7 and most of the supplementary figures.

6. The model figure (Fig. 7A) presents a feedforward loop between Wnt signaling and BX-C genes as definitive, without marking hypothetical steps. Given the correlative nature of the data, the model should be revised to depict hypothesized interactions using dashed lines or similar conventions, and the figure legend should clarify where direct evidence is lacking.

Response: We have revised the Fig. 7A by adding question marks, in addition to the dashed lines, to more clearly indicate the components for which direct evidence is currently lacking. The figure legend has also been revised to reflect these changes and to clarify the speculative nature of these elements.

7. An alternative explanation for the observed adipocyte regionalization is that differences in lipid mobilization and gene expression arise from systemic physiological gradients or differential adipocyte survival, rather than being directly due to *abd-A/Abd-B* expression. This possibility is not considered in the manuscript. The authors should explicitly discuss alternative interpretations, particularly in the context of their data in Figures 1, 2, and 7.

Response: Thank you for this interesting idea. However, we are uncertain about the intended meaning of “systemic physiological gradients” in this context. If such gradients are proposed to underlie the observed regionalization of adipocytes, we would appreciate further clarification on how this alternative model accounts for the full range of experimental findings presented in our study. The model we present, summarized in Fig. 7A, reflects our best effort to integrate all the experimental data, rather than selectively interpreting individual results. While we remain open to alternative interpretations, we believe that any proposed model must explain the comprehensive dataset as effectively as, or more so than, our current framework.

In this regard, we note that *abd-A* and *Abd-B* expression in larval fat body does not follow a smooth anteroposterior gradient. Instead, their expression is sharply delineated – present (“on”) in most of the abdominal adipocytes and absent (“off”) in thoracic adipocytes. We have also obtained additional evidence indicating that *abd-A* and *Abd-B* are transcriptionally regulated in larval fat body. These findings will be detailed in a forthcoming manuscript. Finally, we did not observe evidence of adipocyte death in the larval fat body, which does not support a model based on “differential adipocyte survival”.

8. The broader mechanistic question of how *abd-A* and *Abd-B* modulate both Wnt-activated and Wnt-repressed target genes remains unresolved. The suggestion of a “permissive role” is intriguing but speculative. No co-factors or chromatin states are identified. Preliminary chromatin profiling (e.g., ATAC-seq or CUT&RUN for histone marks) in BX-C-depleted versus control adipocytes could strengthen this model and help interpret the RNA-seq data shown in Figs. 3 and 5 more mechanistically.

Response: We appreciate your insightful suggestions. We agree that this study alone cannot resolve the molecular mechanisms by which *Abd-A* and *Abd-B* modulate Wnt target gene expression. Elucidating these mechanisms would require significant additional resources and is beyond the current scope of this work.

Drosophila larval adipocytes pose unique technical challenges: they are polyploid, limited in number (2100~2500 adipocytes per larva), and densely packed with lipid droplets. These features hinder the application of high-throughput chromatin profiling methods, such as ATAC-seq or CUT&RUN, for studying histone modifications. Additionally, the wide variety of histone marks and uncertainty regarding which are functionally relevant further complicate the experimental design.

As noted earlier, we attempted CUT&RUN using both dissected larval fat bodies and purified nuclei from *pan/dTCF^{EGFP}* larval fat bodies. However, these preparations yielded suboptimal data due to high noise-to-signal ratios, likely caused by the abundance of lipid droplets, and necessitated further optimization. We also performed CUT&RUN analyses with *Abd-A^{EGFP}* and *Abd-B^{EGFP}* in larval brains, where these factors are robustly

expressed in the larval ventral nerve cord (VNC) but not in wing discs (Duckhorn et al., 2022; https://link.springer.com/protocol/10.1007/978-1-0716-2321-3_1). Unfortunately, these experiments also suffered from poor data quality due to a high background signal. While we recognize the importance of understanding chromatin dynamics and potential epigenetic regulation by Abd-A and Abd-B in Wnt target gene expression, addressing these questions will require future methodological advances and dedicated resources.

Moreover, the suggestion to perform chromatin profiling assumes that Abd-A/Abd-B modulate Wnt target gene expression through epigenetic mechanisms. However, alternative models remain possible. For instance, direct interactions between Wnt signaling components Pan-Arm and BX-C proteins have been reported: Abd-A and Abd-B both interact with Pan in bimolecular fluorescence complementation (BiFC) assays (Bischof et al., 2018/PMID: 30247122; Baëza et al., 2015/PMID: 25869471); and Arm interacts with Ubx in a yeast two-hybrid assay (Hsiao et al., 2014/PMID: 25286318). Although it remains unclear whether and how such interactions contribute to the permissive role of Abd-A/Abd-B in regulating Wnt target gene expression, these mechanisms may operate independently of chromatin modifications.

Fig. R5 Bar charts showing mRNA levels of *abd-A*, *Abd-B* and *Axn*. (A, B) Simultaneous depletion of *abd-A* and *Abd-B* in the abdominal fat body results in a 40~60% reduction in their transcript levels, as determined by RNA-seq (A) and RT-qPCR (B). (C) RNA-seq analysis of *Axn* mRNA levels in thoracic and abdominal adipocytes following *Axn* knockdown. Genotypes are color-coded and indicated below each chart. *N* = 3 independent biological replicates. Statistical significance: *p*<0.05 (*), *p*<0.01 (**), and *p*<0.001 (***)

In summary, while we agree that the underlying molecular mechanisms are of great interest, their investigation lies beyond the current scope due to technical and conceptual challenges. We expect that future research will be necessary to address these important questions.

Minor Points:

1. The Introduction section somewhat overstates the novelty of the study, minimizing prior work that has shown BX-C expression and function in the larval fat body (e.g., Duffraisse et al., 2020). Although the authors cite prior studies documenting BX-C gene expression and functional roles in the larval fat body, they do not adequately acknowledge the extent of this earlier work. As a result, the manuscript presents BX-C involvement in fat body biology as a novel discovery, thereby overstating the originality of their findings.

Response: We appreciate your helpful comments and have revised the manuscript with particular attention to our descriptions of both our work and the previous studies by Duffraisse et al., 2020 and others.

2. The discussion speculates extensively on the evolutionary origins of adipocyte heterogeneity, comparing *Drosophila* fat body cells to mammalian white and brown adipocytes (Discussion, final pages). These are interesting ideas but should be explicitly framed as hypotheses rather than conclusions.

Response: We agree with your insightful comments regarding these speculations, and we have revised the manuscript and present these points in a more concise and focused manner.

3. RNAi knockdown efficiencies for *abd-A*, *Abd-B*, and *Axn* are not validated experimentally. Given that many of the central claims rest on RNAi-mediated depletion (e.g., Figs. 2, 4, and 5), the authors should provide qPCR or Western blot quantification of knockdown efficiency.

Response: We generated bar graphs to illustrate the knockdown efficiencies of *abd-A*, *Abd-B*, and *Axn* (Fig. R5). Depletion of *abd-A* and *Abd-B* resulted in over 40% reduction in transcript levels based on RNA-seq analysis (Fig. R5A), and approximately 50-60% reduction as measured by RT-qPCR (Fig. R5B). RNA-seq analysis also confirmed efficient depletion of *Axn* in both thoracic and abdominal fat bodies (Fig. R5C). We note that RNA-seq offers a more quantitative and comprehensive assessment of mRNA transcript abundance compared to traditional qRT-PCR.

Regarding the suggestion to validate RNAi knockdown efficiencies (used in Figs. 2, 4, and 5) by Western blotting for Abd-A, Abd-B, and Axn: both Abd-A and Abd-B proteins are not ubiquitously expressed during the larval stage, as supported by our data (e.g., Fig. 6 and Suppl. Figs. S3, S5, and S13). A single larva has about 2100-2500 adipocytes. Performing Western blotting on dissected larval fat body would require substantial effort to collect sufficient tissue, and this is further complicated by the limited specificity of available antibodies against BX-C proteins, as discussed in both the manuscript and this rebuttal.

For Axn, we have previously confirmed substantial reductions in both protein and mRNA levels in *Axn*¹²⁷ mutant larvae (see Suppl. Fig. S3 in Zhang et al., 2017/PMID: 28827348). Given the extensive analyses presented in our prior studies (Zhang et al., 2017/PMID: 28827348; Liu et al., 2024/PMID: 38968125) and the current work, the precise extent of *Axn* protein reduction (e.g., 40% vs. 70%) in *Axn*^{RNAi} larvae would not affect the core conclusions of these studies.

4. The number of biological replicates used in imaging (e.g., for Figures 2, 4, and 5) and for quantifications (e.g., Figure 5H) should be clearly indicated in each figure legend.

Response: We have revised the figure legends to include this information in the corresponding figures.

5. The manuscript describes statistical analysis methods only briefly. One-tailed t-tests are used throughout without proper justification. Unless the authors can rigorously justify the use of one-tailed tests based on experimental design, standard two-tailed t-tests should be employed.

Fig. R6 Re-analyses of the data shown in Fig. 5H using a two-tailed t-test. The differences remain statistically significant compared to the one-tailed t-tests presented in Fig. 5H. Statistical significance: $p < 0.05$ (*/#), $p < 0.01$ (**/##), and $p < 0.001$ (***/###). Asterisks (*) denote comparisons with the '*dCg-Gal4/+*' control, while pound signs (#) indicate comparisons with '*dCg-Gal4/Axn*^{RNAi}, +' (small cells).

Response: Thank you for raising this point. The primary difference between one-tailed and two-tailed t-tests lies in the nature or directionality of the hypothesis being tested. A one-tailed t-test evaluates whether there is a statistically significant effect in a specific direction (e.g., an increase or a decrease), whereas a two-tailed t-test assesses for any difference, regardless of direction. One-tailed t-tests provide greater statistical power when there is a strong, directional hypothesis, while two-tailed t-tests are more conservative and suitable when changes in either direction are plausible. In our case, the experimental models are based on well-defined directional predictions regarding Wnt signaling outcomes, which justifies the use of one-tailed t-tests in our primary analyses. Nevertheless, to address this concern, we re-analyzed the data using two-tailed t-tests and found that the results remain statistically significant (**Fig. R6**).

6. Supplementary figures, such as the validation of EGFP-tagged lines (Supplementary Figs. S4-S5), are important for interpreting the main data but are not sufficiently discussed in the text. Clear integration of validation data into the Results section would improve clarity.

Response: Thank you for pointing this out. We have revised the manuscript to include a more detailed description of these supplementary data.

Conclusion: Major Revisions

While the manuscript presents interesting observations regarding regional specialization of adipocytes and an interaction between Wnt signaling and Hox gene expression, the study lacks direct mechanistic proof, and the interpretation of the data exceeds the strength of the evidence. Major experimental revisions - including direct testing of Wnt regulation of BX-C genes in adipocytes, rigorous phenotypic quantification, broader Wnt reporter validation, sample purity assessment, and a more cautious presentation of the model - are required for the manuscript to meet the standards expected at EMBO Reports. I therefore recommend major revisions.

Response: We have revised the manuscript and updated the figures in accordance with your insightful suggestions. We hope that the revised manuscript meets the high standards of *EMBO Reports*.

Dear Ji,

Thank you for submitting your revised manuscript. It has now been seen by both of the original referees.

As you will see, referees find that the study is significantly improved during revision and recommend publication. However, both referees have remaining minor outstanding concerns. Referee #1 would like further explanation on the different wnt sensitivity of posterior and thoracic cells (point 1). Moreover, referee #1 raises a point regarding the proposed activation of posterior *hox* genes by wnt signaling (point 2). Referee #2 has numerous minor concerns to be addressed as per his/her suggestions as well. Please address all concerns textually by adding discussion points and as per referee recommendations. Please provide a point-by-point response. Please let me know if you would like to discuss any of the points further.

Moreover, the editorial points below need to be addressed before I can accept the manuscript.

- Please make the dataset GSE280511 publicly available and remove the referee token from the manuscript text. Please add a URL, which directly resolves to the dataset.
- Please rename the Data availability statement section as Data Availability.
- Please rename the Competing interests section as Disclosure and Competing Interests Statement.
- Please remove the Author contributions section from the manuscript text.
- Please fill out and include an author checklist as listed in our online guidelines (<https://www.embopress.org/page/journal/14693178/authorguide>).
- We note the presence of a wrong callout, Fig. S1A, which needs to be corrected.
- We note the following regarding the Appendix file: Appendix figure and table legends need to be removed from the manuscript file. Appendix Figure 10 in the legend of this figure should be corrected to Appendix Figure S10. Appendix Figure 12 in the legend of this figure should be corrected to Appendix Figure S12.
- All research articles submitted as revised versions must include a structured methods section that includes a Reagents and Tools Table followed by a Methods and Protocols section. Please see <https://www.embopress.org/page/journal/14693178/authorguide#structuredmethods> for further information.
- We note that source data for Figures 4AB, 5ABC, 6A, 7CDEF were not provided, which we are aware could be a part of GSE280511. In which case, please include this piece of information in the source data checklist.
- The manuscript sections should be in the following order: Title page - Abstract & Keywords - Introduction - Results - Discussion - Methods - Data Availability - Acknowledgments - Disclosure Statement & Competing Interests - References - Figure Legends - (Main Tables with legends if applicable) - Expanded View Figure Legends.
- During our routine figure checks, we were unable to detect background signal in the following panels: Appendix Fig S3 B' C' E' F'. Please provide source data for these panels.
- Our production/data editors have asked you to clarify several points in the figure legends - Figure Legends (main + EV):
 - o Please define the annotated p values ****/**/**/* as well as provide the exact p-values for the same in the legend of figure 2P, 4G, H as appropriate.
 - o Please note that the exact p values are not provided in the legend of figure 5H.
 - o Please indicate the statistical test used for data analysis in the legends of figures 2P; 3A, 4A, G, H.
 - o Please note that information related to n is missing in the legend of figure 4H.
 - o Please note that the error bars are not defined in the legends of figures 2P, 4G, H; 5H.
 - o Please note that the scale bar is missing for figures 1A-C, K, K', L-L', M, M', N, N', O, O', P; 3J, K, N; 4C, C', D, D', E, E', F; 5D, D', E, E', F, F', G; 6B-E', F'-G'; 6H-I', J'-K'.
 - o Please note that scale bar and its definition are missing for figure 2A.
 - o Please note that the dotted borders are not defined in the legend of figures 5E, E'. This needs to be rectified.
- Papers published in EMBO Reports include a 'synopsis' and 'bullet points' to further enhance discoverability. Both are displayed on the html version of the paper and are freely accessible to all readers. The synopsis includes a short standfirst summarizing the study in 1 or 2 sentences (max 35 words) that summarize the paper and are provided by the authors and streamlined by the handling editor. I would therefore ask you to include your synopsis blurb and 3-5 bullet points listing the key experimental findings.
- In addition, please provide an image for the synopsis. This image should provide a rapid overview of the question addressed in the study but still needs to be kept fairly modest since the image size cannot exceed 550 (width) x 300-600 (height) pixels.

Thank you again for giving us to consider your manuscript for EMBO Reports, I look forward to your minor revision.

Kind regards,

Deniz

--

Deniz Senyilmaz Tiebe, PhD
Senior Scientific Editor

Referee #1:

I thank the authors of " Adipocyte heterogeneity regulated by the Bithorax Complex-Wnt signaling crosstalk in Drosophila" for their thorough responses to my questions. In this manuscript, the authors show that inducing wnt signaling causes abdominal adipocytes to metabolize lipids. This requires the posterior hox proteins Abd-A and Abd-B, which are expressed in the abdominal fat body cells but not the thoracic. The authors provide a lot of data showing that the genes involved in this break down are miss-regulated after wnt signaling and that these misregulations require the hox proteins. Overall, I find the data presented here to be interesting and plentiful. I still have a two main questions regarding this manuscript.

1. My major concern with the manuscript centers around the biological significance. When does this happen and why does the fly spend so much effort to make the posterior cells wnt responsive but not the thoracic? Based on what the authors say, there is little or no wnt signaling in the fat body at the time of the experiments. In fact, the authors state that their ftz3 reporter is not activated in control samples. If the authors remove abd-A and Abd-B, they can suppress wnt signaling in the posterior FB, yet they only get a one day delay in development. This seems to be unrelated to the wnt signaling as other mutants that impair wnt signaling do not share this phenotype. So, as I asked in my previous review, "is this of developmental importance or is this just explaining how ectopic activation of wnt signaling causes a phenotype?" I ask this only to put this work into perspective. If the authors could show that wnt initiation of abdominal lipid metabolism is important for something in the normal life of the fly, it would skyrocket the value of this work.

2. The authors believe that wnt signaling activates the posterior hox genes. But this is limited to the cells that already express the posterior hox genes. I have some questions about this. If one overexpresses the posterior hox genes in the thoracic FB, the authors show that the thoracic FB cells behave like the abdominal cells. How does this happen? Using the ftz reporter, the authors never seem (that I found) to see expression in the thorax. Do these cells express the genes necessary for wnt signaling to happen? Might the hox genes activate wnt signaling genes? It just seem like this would be an easier way to get these effects, though biology does not always work via the easiest path.

In closing, I would like to say that I found the work here well done and well presented. I am just not sure about the biological impact of this work. If the editors and my fellow reviewers do not share my concerns, I would not be against this manuscript being published in EMBO reports.

Referee #2:

Review for EMBOR-2025-61647V2

Hemba-Waduge et al. examine regional adipocyte heterogeneity in Drosophila and the crosstalk between Wnt signaling and BX-C (abd-A/Abd-B). The revision substantially strengthens the case with (i) added Wnt readouts (quantified fz3-RFP; RT-qPCR for CycD, nkd, Notum), (ii) HCR-RNA FISH for fz3/FASN1/Lsd-2, (iii) adipocyte-nuclei CUT&RUN for dTCF/Pan at abd-A/Abd-B (alongside wing-disc tracks), and (iv) a revised model that explicitly marks speculative steps; several requested quantifications are now provided (adipocyte size; TGs) and figure legends include replicate information. Collectively, the data now support a permissive role for abd-A/Abd-B in Wnt-responsive transcription in abdominal adipocytes, with Wnt further reinforcing abd-A/Abd-B expression.

Major Points (remaining to address)

1. Autonomy/temporal control. The response declines Gal80ts and does not add clonal tests, arguing temperature confounds; this remains a limitation for excluding systemic effects. Please state this explicitly in the Discussion (cell-autonomous claims should be framed cautiously), or add a strictly cell-autonomous assay.
2. Statistics and units of replication. The main text still declares "one-tailed unpaired t-tests ... minimum of three independent biological replicates" globally; Fig. 4G specifies three biological replicates, whereas Fig. 5H reports per-cell counts (4-23 cells) without clearly stating the biological n. Either (i) standardize to two-tailed tests in the main figures (the rebuttal shows significance is retained for Fig. 5H under two-tailed tests) and specify the biological unit and n per panel, or (ii) provide a brief, figure-specific justification for one-tailed tests.
3. Direct Wnt→BX-C input. The addition of adipocyte CUT&RUN (S12) is valuable, but the authors also acknowledge high noise; several statements still read as definitive. Please keep wording uniformly cautious (e.g., "supports"/"may directly regulate") and point readers to the high background noted for adipocyte CUT&RUN.
4. RNA-seq sample purity. The FlyAtlas2 cross-check (R4) is helpful; to make this durable, move the essential purity analysis (R4) into the Supplement and cross-reference it in Results/Methods alongside the dissection schematic that flags ovary position (Appendix Fig. S1H).
5. Knockdown validation. The rebuttal provides RNA-seq/RT-qPCR-based knockdown estimates for abd-A/Abd-B/Axn (R5). Please include these plots in the Supplement and cite them in the relevant Results sections.
6. Bring rebuttal-only quantifications into the paper. Adipocyte size (R1-R2) and TG (R3) quantifications directly address review

requests; these should appear in the Supplement (with full legends and statistics) and be cited from the main text.

Minor Points

1. Model figure legend has a typo ("mechamism"); please correct.
2. Figure naming is inconsistent ("Appendix Figure 12" vs "Appendix Figure S12"); standardize.
3. Ensure each figure legend explicitly states the biological n and the statistical test used (and whether bars are SD or SEM) to match the "minimum of three biological replicates" statement in Methods.
4. Where Abd-B effects are shown only in the appendix (e.g., S10), add a brief pointer from the main text.

Conclusion: Minor Revision

The revision addresses the core of the prior concerns with new mechanistic and quantitative data (RT-qPCR for Wnt targets; fz3/FASN1/Lsd-2 HCR; adipocyte-nuclei CUT&RUN; improved model). With the editorial/organizational fixes above, especially standardizing statistics/reporting, tempering language around adipocyte CUT&RUN, and incorporating rebuttal-only quantifications/knockdown validation into the Supplement, the manuscript meets EMBO Reports' bar.

We sincerely thank the reviewers for their constructive and thoughtful feedback, and we provide our detailed point-by-point responses below.

Referee #1: I thank the authors of "Adipocyte heterogeneity regulated by the Bithorax Complex-Wnt signaling crosstalk in *Drosophila*" for their thorough responses to my questions. In this manuscript, the authors show that inducing wnt signaling causes abdominal adipocytes to metabolize lipids. This requires the posterior hox proteins Abd-A and Abd-B, which are expressed in the abdominal fat body cells but not the thoracic. The authors provide a lot of data showing that the genes involved in this break down are miss-regulated after wnt signaling and that these misregulations require the hox proteins. Overall, I find the data presented here to be interesting and plentiful. I still have a two main questions regarding this manuscript.

1. My major concern with the manuscript centers around the biological significance. When does this happen and why does the fly spend so much effort to make the posterior cells wnt responsive but not the thoracic? Based on what the authors say, there is little or no wnt signaling in the fat body at the time of the experiments. In fact, the authors state that their *ftz3* reporter is not activated in control samples. If the authors remove *abd-A* and *Abd-B*, they can suppress wnt signaling in the posterior FB, yet they only get a one day delay in development. This seems to be unrelated to the wnt signaling as other mutants that impair wnt signaling do not share this phenotype. So, as I asked in my previous review, "is this of developmental importance or is this just explaining how ectopic activation of wnt signaling causes a phenotype?" I ask this only to put this work into perspective. If the authors could show that wnt initiation of abdominal lipid metabolism is important for something in the normal life of the fly, it would skyrocket the value of this work.

2. The authors believe that wnt signaling activates the posterior hox genes. But this is limited to the cells that already express the posterior hox genes. I have some questions about this. If one overexpresses the posterior hox genes in the thoracic FB, the authors show that the thoracic FB cells behave like the abdominal cells. How does this happen? Using the *ftz* reporter, the authors never seem (that I found) to see expression in the thorax. Do these cells express the genes necessary for wnt signaling to happen? Might the hox genes activate wnt signaling genes? It just seem like this would be an easier way to get these effects, though biology does not always work via the easiest path.

In closing, I would like to say that I found the work here well done and well presented. I am just not sure about the biological impact of this work. If the editors and my fellow reviewers do not share my concerns, I would not be against this manuscript being published in EMBO reports.

Response: Thank you for your insightful and constructive comments.

Regarding the biological significance of adipocyte heterogeneity: This is indeed a very intriguing question. Our study was driven by curiosity, and our results suggest that regional heterogeneity of the larval fat body is not a stochastic event but rather is controlled by the genetic circuitry we describe. Although we do not yet have a definitive answer, we propose several possible explanations (as outlined in the manuscript and our previous rebuttal) that warrant future investigation: (**a**) Adipocyte heterogeneity has also been reported in other insect species, and its potential significance has been discussed by Haunerland and Shirk (*Annual Review of Entomology* 40, 121-145, 1995); (**b**) subtle differences among adipocytes may confer adaptive advantages under stress conditions such as starvation, nutrient fluctuations, temperature shifts, or bacterial infection, analogous to the distinct functions of brown, white, and beige adipocytes in mammals. Since all of our analyses were performed under standard laboratory conditions, it will be important to examine whether regional differences of the larval fat body become more apparent under physiological perturbations or pathological challenges; (**c**) from an evolutionary perspective, the diversification of adipocyte types may not have emerged suddenly, but could have deeper roots in more primitive animals such as insects; and (**d**) thoracic adipocytes in wild-type larvae exhibit strong kynurenine autofluorescence relative to abdominal adipocytes, suggesting that, because tryptophan and kynurenine are precursors of the eye pigment ommochrome and thoracic adipocytes lie closer to the eye imaginal discs, they may facilitate shuttling of these metabolites to the discs during metamorphosis. We further speculate that the crosstalk between Wnt signaling and *HOX* genes may contribute to regional differences among adipose depots in mammals. If so, dissecting these mechanisms in the simpler *Drosophila* model will provide valuable evolutionary and mechanistic insights.

Regarding the comment that "the authors believe that wnt signaling activates the posterior hox genes": Our results indicate that Wnt signaling *potentiates*, rather than *activates*, the expression of *abd-A* and *Abd-B*. This distinction is important. While *abd-A* and *Abd-B* are expressed in most abdominal adipocytes

and active Wnt signaling further enhances their transcription, these genes are undetectable in thoracic adipocytes, and Wnt signaling alone is insufficient to induce their transcription *de novo*. Instead, intermediate states may exist: some thoracic adipocytes could express very low levels of *abd-A* and *Abd-B*, or their chromatin may not be fully “locked” against transcription. Because there is no strict boundary between the thoracic and abdominal regions, Wnt signaling in such thoracic adipocytes could stimulate *abd-A* and *Abd-B* expression, which in turn amplify Wnt signaling, ultimately enabling lipid mobilization responses similar to those observed in abdominal adipocytes (as shown in Fig. 2E/E’ and 2G/G’). This interpretation is further supported by our observation that *Axn*¹²⁷ mutant larvae display stronger effects in larval adipocyte heterogeneity than fat body-specific depletion of *Axn* via RNAi (e.g., Fig. 1B cf. Fig. 1D). We also agree with your remark that “biology does not always work via the easiest path.”

Referee #2:

Review for EMBOR-2025-61647V2

Hemba-Waduge et al. examine regional adipocyte heterogeneity in *Drosophila* and the crosstalk between Wnt signaling and BX-C (*abd-A/Abd-B*). The revision substantially strengthens the case with (i) added Wnt readouts (quantified *fz3*-RFP; RT-qPCR for *CycD*, *nkd*, *Notum*), (ii) HCR-RNA FISH for *fz3/FASN1/Lsd-2*, (iii) adipocyte-nuclei CUT&RUN for dTCF/Pan at *abd-A/Abd-B* (alongside wing-disc tracks), and (iv) a revised model that explicitly marks speculative steps; several requested quantifications are now provided (adipocyte size; TGs) and figure legends include replicate information. Collectively, the data now support a permissive role for *abd-A/Abd-B* in Wnt-responsive transcription in abdominal adipocytes, with Wnt further reinforcing *abd-A/Abd-B* expression.

Response: Thank you for your insightful comments and for carefully reviewing our manuscript. These are excellent suggestions, and we have revised the manuscript accordingly.

Major Points (remaining to address)

1. Autonomy/temporal control. The response declines Gal80ts and does not add clonal tests, arguing temperature confounds; this remains a limitation for excluding systemic effects. Please state this explicitly in the Discussion (cell-autonomous claims should be framed cautiously), or add a strictly cell-autonomous assay.

Response: We have revised the manuscript accordingly by explaining this further in the discussion.

2. Statistics and units of replication. The main text still declares “one-tailed unpaired t-tests ... minimum of three independent biological replicates” globally; Fig. 4G specifies three biological replicates, whereas Fig. 5H reports per-cell counts (4-23 cells) without clearly stating the biological n. Either (i) standardize to two-tailed tests in the main figures (the rebuttal shows significance is retained for Fig. 5H under two-tailed tests) and specify the biological unit and n per panel, or (ii) provide a brief, figure-specific justification for one-tailed tests.

Response: We have revised the manuscript according to your suggestions by providing a brief, figure-specific justification for one-tailed tests.

3. Direct Wnt→BX-C input. The addition of adipocyte CUT&RUN (S12) is valuable, but the authors also acknowledge high noise; several statements still read as definitive. Please keep wording uniformly cautious (e.g., “supports”/“may directly regulate”) and point readers to the high background noted for adipocyte CUT&RUN.

Response: We have revised the manuscript according to your suggestions.

4. RNA-seq sample purity. The FlyAtlas2 cross-check (R4) is helpful; to make this durable, move the essential purity analysis (R4) into the Supplement and cross-reference it in Results/Methods alongside the dissection schematic that flags ovary position (Appendix Fig. S1H).

Response: We have revised the manuscript according to your suggestions.

5. Knockdown validation. The rebuttal provides RNA-seq/RT-qPCR-based knockdown estimates for *abd-*

A/Abd-B/Axn (R5). Please include these plots in the Supplement and cite them in the relevant Results sections.

Response: We have revised the manuscript according to your suggestions.

6. Bring rebuttal-only quantifications into the paper. Adipocyte size (R1-R2) and TG (R3) quantifications directly address review requests; these should appear in the Supplement (with full legends and statistics) and be cited from the main text.

Response: We have revised the manuscript according to your suggestions.

Minor Points

1. Model figure legend has a typo ("mechamism"); please correct.

Response: We have corrected the typo in the model figure legend.

2. Figure naming is inconsistent ("Appendix Figure 12" vs "Appendix Figure S12"); standardize.

Response: We have corrected the typo in the manuscript.

3. Ensure each figure legend explicitly states the biological n and the statistical test used (and whether bars are SD or SEM) to match the "minimum of three biological replicates" statement in Methods.

Response: All bars are SD but not SEM. We have revised the manuscript according to your suggestions.

4. Where Abd-B effects are shown only in the appendix (e.g., S10), add a brief pointer from the main text.

Response: We have revised the manuscript according to your suggestions.

Conclusion: Minor Revision

The revision addresses the core of the prior concerns with new mechanistic and quantitative data (RT-qPCR for Wnt targets; fz3/FASN1/Lsd-2 HCR; adipocyte-nuclei CUT&RUN; improved model). With the editorial/organizational fixes above, especially standardizing statistics/reporting, tempering language around adipocyte CUT&RUN, and incorporating rebuttal-only quantifications/knockdown validation into the Supplement, the manuscript meets EMBO Reports' bar.

Response: We sincerely thank you again for your careful review. We have revised the manuscript according to your suggestions.

Dr. Jun-Yuan Ji
Tulane University School of Medicine
Biochemistry and Molecular Biology
Louisiana Cancer Research Center
1700 Tulane Avenue
New Orleans, LA 70112
United States

Dear Dr. Ji,

I am pleased to inform you that your manuscript has been accepted for publication in EMBO reports. Your manuscript will be processed for publication by EMBO Press. It will be copy edited and you will receive page proofs prior to publication. Please note that you will be contacted by Springer Nature Author Services to complete licensing and payment information.

Yours sincerely,

Kurt Weir
Editor
EMBO Reports
